# Singular Subspace Perturbation Bounds via Rectangular Random Matrix Diffusions

**Peiyao Lai**
Worcester Polytechnic Institute
Worcester, MA, USA

**Oren Mangoubi**
Worcester Polytechnic Institute
Worcester, MA, USA

## Abstract

Given a matrix $A \in \mathbb{R}^{m \times d}$ with singular values $\sigma_1 \geq \cdots \geq \sigma_d$, and a random matrix $G \in \mathbb{R}^{m \times d}$ with iid $N(0, T)$ entries for some $T > 0$, we derive new bounds on the Frobenius distance between subspaces spanned by the top-$k$ (right) singular vectors of $A$ and $A + G$. This problem arises in numerous applications in statistics where a data matrix may be corrupted by Gaussian noise, and in the analysis of the Gaussian mechanism in differential privacy, where Gaussian noise is added to data to preserve private information. We show that, for matrices $A$ where the gaps in the top-$k$ singular values are roughly $\Omega(\sigma_k - \sigma_{k+1})$ the expected Frobenius distance between the subspaces is $\tilde{O}(\frac{\sqrt{d}}{\sigma_k - \sigma_{k+1}} \times \sqrt{T})$, improving on previous bounds by a factor of $\frac{\sqrt{m}}{\sqrt{d}}$. To obtain our bounds we view the perturbation to the singular vectors as a diffusion process– the Dyson-Bessel process– and use tools from stochastic calculus to track the evolution of the subspace spanned by the top-$k$ singular vectors, which may be of independent interest.

## 1 Introduction

Given a matrix $A \in \mathbb{R}^{m \times d}$ with $d \leq m$ and singular values $\sigma_1 \geq \cdots \geq \sigma_d$, one oftentimes wishes to approximate the right-singular vectors of $A$ by a lower rank matrix of some rank $k < d$ Shikhaliev et al. (2019); Hubert & Engelen (2004); James et al. (2013); Kishore Kumar & Schneider (2017); Liberty et al. (2007). For instance, one may wish to learn the subspace spanned by the top-$k$ right-singular vectors of $A$, in which case one may seek a projection matrix which minimizes the distance to the projection matrix onto the subspace spanned by the top-$k$ right-singular vectors of $A$. One can also consider the related problem of recovering a matrix $\hat{M}_k$ which minimizes the Frobenius distance $\|\hat{M}_k - A^\top A\|_F$ to the covariance matrix $A^\top A$ of the data $A$. Roughly speaking, both of these problems are instances of the following general problem: given a set of numbers $\gamma_1 \geq \cdots \geq \gamma_d$ and denoting by $\Gamma := \mathrm{diag}(\gamma_1, \cdots, \gamma_d)$ and by $A = U \Sigma V^\top$ a singular value decomposition of $A$, find a matrix $M \in \mathcal{O}_{\Gamma^2}$ which minimizes the Frobenius distance $\|M - V^\top \Gamma^2 V\|_F$, where $\mathcal{O}_{\Gamma^2} := \{U \Gamma^2 U^\top : U \in O(d)\}$ denotes the orbit of $\Gamma^2$ under the orthogonal group. Plugging in $\gamma_i = 1$ for $i \leq k$ and $\gamma_i = 0$ for $i > k$, we recover the problem of finding a projection matrix which minimizes the Frobenius distance to the projection matrix onto the subspace spanned by to top-$k$ right-singular vectors of $A$. And, roughly speaking, when we set $\gamma_i \approx \sigma_i$ for $i \leq k$ and $\gamma_i = 0$ for $i > k$, we recover the problem of finding a rank-$k$ covariance matrix which minimizes the Frobenius distance to the covariance matrix of $A$.

In many applications, the matrix $A$ is perturbed by a "noise" matrix $E \in \mathbb{R}^{m \times d}$ and one only has access to a perturbed matrix $A + E$. Oftentimes, the noise matrix consist of iid Gaussian entries. For instance, in statistics applications, and signal and image processing applications, this noise may arise as natural background Gaussian noise obscuring a "signal" matrix $A$ Wu & Chen (1997); Helstrom (1955); Liu & Lin (2012); DjuriC (1996); Bergmans (1974). In differential privacy applications, Gaussian noise may be artificially added to the data matrix $A$, or to a machine learning algorithm trained on the data $A$, to hide sensitive information about individuals in the dataset Dwork (2006); Dwork et al. (2006); see e.g. Dwork et al. (2014); Mangoubi & Vishnoi (2022; 2023; In Press) where *symmetric*-matrix Gaussian noise is added to covariance matrices to guarantee privacy. The addition

of Gaussian noise to ensure privacy is referred to as the Gaussian mechanism, and is known to satisfy $(\varepsilon, \delta)$-differential privacy guarantees.

## 1.1 RELATED WORK

Multiple prior works have shown singular subspace perturbation bounds when $E \in \mathbb{R}^{m \times d}$ may be any (deterministic) matrix. For instance, the Davis-Kahan-Wedin sine-Theta theorem Davis & Kahan (1970); Wedin (1972) implies a bound of roughly

$$|||V_k V_k^\top - \hat{V}_k \hat{V}_k^\top||| \leq \frac{|||E|||}{\sigma_k - \sigma_{k+1}}, \tag{1}$$

where $V_k, \hat{V}_k$ are, respectively, the matrices whose columns are the top-$k$ right-singular vectors of $A$ and $\hat{A} := A + E$, and $||| \cdot |||$ is e.g. the Frobenius norm $\| \cdot \|_F$ or the spectral norm $\| \cdot \|_2$. These bounds are tight (for sufficiently small $|||E|||$) in the general setting where $E \in \mathbb{R}^{m \times d}$ may be any (deterministic) matrix.

When the perturbation $E$ is, e.g., a Gaussian random matrix with iid $N(0, T)$ entries for some $T > 0$, one can plug in high-probability concentration bounds, which imply that $\|E\|_2 \leq O(\sqrt{m})$ w.h.p., to the deterministic bounds in equation 1 to obtain a bound of

$$\|V_k V_k^\top - \hat{V}_k \hat{V}_k^\top\|_F \leq \frac{\sqrt{k}\sqrt{m}}{\sigma_k - \sigma_{k+1}} \times \sqrt{T}$$

w.h.p. However, the resulting bounds may not be tight.

Multiple works have obtained tighter bounds than those implied by the deterministic bounds in equation 1, in different settings when $E$ is a random matrix from some known distribution or class of distributions (see e.g. O'Rourke et al. (2018); Fan et al. (2018); Abbe et al. (2022); Cai et al. (2021)). In particular, when their bounds, which are given in terms of the spectral norm, are applied to bounding the Frobenius norm distance, the results in O'Rourke et al. (2018) imply that if the entries of $E$ satisfy concentration properties which generalize those of Gaussian distributions, and $A$ has rank $r \leq d$, then

$$\|\hat{V}_k \hat{V}_k^\top - V_k V_k^\top\|_F \leq O\left(k\left(\frac{\sqrt{r}}{\sigma_k - \sigma_{k+1}} + \frac{m}{\sigma_k(\sigma_k - \sigma_{k+1})} + \frac{\sqrt{m}}{\sigma_k}\right)\right) \tag{2}$$

w.h.p. In O'Rourke et al. (2023) the authors obtain singular subspace perturbation bounds when $E$ is a random matrix with iid standard Gaussian entries. Their results, which are given as bounds on the spectral norm, imply that $\max\left(\|\hat{U}_k \hat{U}_k^\top - U_k U_k^\top\|_2, \ \|\hat{V}_k \hat{V}_k^\top - V_k V_k^\top\|_2\right) \leq O\left(r\sqrt{\sum_{j=1}^{k} \frac{1}{(\sigma_j - \sigma_{k+1})^2}} + \frac{\sqrt{m}\sqrt{k}}{\sigma_k}\right)$. This in turn implies bounds on the Frobenius norm of

$$\max\left(\|\hat{U}_k \hat{U}_k^\top - U_k U_k^\top\|_F, \ \|\hat{V}_k \hat{V}_k^\top - V_k V_k^\top\|_F\right) \leq O\left(r\sqrt{k}\sqrt{\sum_{j=1}^{k} \frac{1}{(\sigma_j - \sigma_{k+1})^2}} + \frac{\sqrt{m}\sqrt{k}}{\sigma_k}\right). \tag{3}$$

O'Rourke et al. (2023) show that their spectral norm bounds are tight with respect to the subspace spanned by the top-$k$ $m$-dimensional *left* singular vectors of $A \in \mathbb{R}^{m \times d}$ when $m \geq d$. However, the bounds in equation 3 do not imply tight bounds on the perturbation $\hat{V}_k \hat{V}_k^\top - V_k V_k^\top$ to the subspace spanned by the top-$k$ $d$-dimensional *right* singular vectors. In particular, the bound on the peturbation $\hat{V}_k \hat{V}_k^\top - V_k V_k^\top$ to the subspace spanned by the top-$k$ $d$-dimensional right singular vectors implied by equation 3 grows proportional to the (square root of) the larger of the matrix dimensions $\sqrt{m}$.

*This leads to the question of whether one can obtain improved bounds on the perturbation $\|\hat{V}_k \hat{V}_k^\top - V_k V_k^\top\|_F$ to the subspace spanned by the top-$k$ $d$-dimensional right singular vectors of an $m \times d$ matrix $A$ with $m > d$ perturbed by Gaussian noise,* which do not grow with the larger dimension $m$.

Subspace perturbation bounds have also been obtained in different settings where the input matrix, and random matrix perturbation, is a *symmetric* matrix (see e.g. Dwork et al. (2014); Eldridge et al. (2018); Fan et al. (2018)). For instance, Dwork et al. (2014) obtain perturbation bounds for covariance

matrices perturbed by symmetric Gaussian noise, and apply these perturbation bounds to a version of the Gaussian mechanism to obtain tighter utility bounds for covariance matrix approximation problems under $(\varepsilon, \delta)$-differential privacy. Mangoubi & Vishnoi (2022; 2023) improve on some of their utility bounds by viewing the addition of the symmetric Gaussian noise as a symmetric-matrix valued stochastic process, and use tools from and stochastic calculus and random matrix theory to bound the perturbation to the symmetric matrix eigenvectors.

## 1.2 OUR CONTRIBUTIONS

Given any matrix $A \in \mathbb{R}^{m \times d}$, and a set of numbers $\gamma_1 \geq \cdots \geq \gamma_d$, our main result (Theorem 2.2) is a bound on the perturbation to the matrix $V^{\top} \Gamma^2 V \in \mathcal{O}_{\Gamma^2}$ where $A = U \Sigma V^{\top}$ is a singular value decomposition of $A$. We show that, if the matrix $A$ is perturbed by a matrix $E = \sqrt{T} G$, where $T > 0$ and $G$ is a Gaussian random matrix with iid $N(0,1)$ entries, the right-singular vectors $\hat{V} = (\hat{v}_1, \cdots, \hat{v}_d)$ of the perturbed matrix $A + \sqrt{T} G$ satisfy the bound

$$\mathbb{E}\left[\|\hat{V}\Gamma^{\top}\Gamma\hat{V}^{\top} - V\Gamma^{\top}\Gamma V^{\top}\|_F\right] \leq O\left(\sqrt{\sum_{i=1}^{k}\sum_{j=i+1}^{d}\frac{(\gamma_i^2 - \gamma_j^2)^2}{(\sigma_i - \sigma_j)^2}}\sqrt{T}\right),$$

where the right-hand-side is a sum-of-squares of the ratios of the eigenvalue gaps of $\Gamma$ and $\Sigma$.

Plugging in different values of $\gamma$, we obtain as corollaries bounds for the subspace recovery and low-rank covariance matrix approximation problems. In particular, show that $\|V_k V_k^{\top} - \hat{V}_k \hat{V}_k^{\top}\|_F \leq O\left(\frac{\sqrt{d}}{\sigma_k - \sigma_{k+1}}\sqrt{T}\right)$ whenever the top-$k$ singular value gaps of $A$ are roughly $\Omega(\max(\sigma_k - \sigma_{k+1}, \sqrt{m}\sqrt{T}))$ (Corollary 2.3). This improves (in expectation) on the bounds implied by both Davis & Kahan (1970); Wedin (1972) and O'Rourke et al. (2018) by a factor of roughly $\frac{\sqrt{m}}{\sqrt{d}}\sqrt{k}$, and by a factor of $\frac{\sqrt{m}}{\sqrt{d}}$ over the bounds implied by O'Rourke et al. (2023), in the above setting where $E$ is a Gaussian random matrix. In particular, our bound replaces those bounds' dependence on the number of *rows* $m$ with the number of *columns* $d$. This can lead to a large improvement in many applications, as one oftentimes has that the number $m$ of rows in the data matrix (corresponding to the number of datapoints) is much larger than the number of columns $d$ (which oftentimes correspond to different features in the data). Our results also imply similar improvements for the low-rank covariance matrix approximation problem (Corollary 2.4).

To obtain our bounds, building on several previous works, including Dyson (1962); Norris et al. (1986); Bru (1989); Mangoubi & Vishnoi (2022; 2023; In Press), we view the perturbation of a matrix $A \in \mathbb{R}^{m \times d}$ by Gaussian noise as a Brownian motion on the entries of an $\mathbb{R}^{m \times d}$ matrix, $\Phi(t) := A + B(t)$ where $B(t)$ is a $m \times d$ matrix whose entries undergo iid standard Brownian motions. This Brownian motion induces a stochastic diffusion process on the singular values and singular vectors of $\Phi(t)$, referred to as the Dyson-Bessel process. The evolution of these eigenvalues and eigenvectors is determined by a system of stochastic differential equations (see e.g. Dyson (1962); Norris et al. (1986); Guionnet & Huang (2021)). This allows us to use Ito's lemma from stochastic calculus to track the evolution of the Frobenius distance as a stochastic integral of a sum-of-squares of perturbations to the (right)-singular vectors of $\Phi(t)$. In particular, the stochastic evolution of the eigenvectors allows us to bypass higher-order matrix derivative terms that arise in Taylor expansions of deterministic perturbations, as these terms vanish in the stochastic derivative when the perturbation is a Brownian motion, due to the independence of random noise additions at each infinitesimal time-step of the Brownian motion. This in turn allows us to obtain stronger bounds than would be possible in the deterministic setting.

## 2 MAIN RESULTS

For any $d > 0$, denote by $\mathrm{O}(d)$ the group of orthogonal of $d \times d$ matrices. For any diagonal matrix $\Lambda \in \mathbb{R}^{d \times d}$, denote by $\mathcal{O}_{\Lambda} := \{U \Lambda U^{\top} : U \in O(d)\}$ the orbit of $\Lambda$ under the orthogonal group.

Given any matrix $A \in \mathbb{R}^{m \times d}$, where $d \leq m$, with singular values $\sigma_1 \geq ... \geq \sigma_d \geq 0$ and corresponding orthonormal right-singular vectors $v_1, ... v_d$, and given any numbers $\gamma_1 \geq \cdots \geq \gamma_d$,

our main result (Theorem 2.2) is a bound on the perturbation to the matrix $V^\top \Gamma^2 V \in \mathcal{O}_{\Gamma^2}$, where $V := [v_1, ... v_d] \in \mathbb{R}^{d \times d}$ and $\Gamma := \mathrm{diag}(\gamma_1, \ldots, \gamma_d)$.

Our main result holds under the following assumption on the gaps in the top $k + 1$ singular values $\sigma_1 \geq ... \geq \sigma_{k+1}$ of the matrix $A$. We note that this assumption is satisfied on many real-world datasets whose singular values exhibit exponential decay (see e.g. Appendix J of Mangoubi & Vishnoi (2022) for examples of datasets with exponentially-decaying singular values).

**Assumption 2.1** $(A, k, T, \sigma, \gamma)$ **(Singular value gaps).** *The gaps in the top $k + 1$ singular values $\sigma_1 \geq ... \geq \sigma_{k+1}$ of the matrix $A \in \mathbb{R}^{m \times d}$ satisfy $\sigma_i - \sigma_{i+1} \geq 8\sqrt{T}\sqrt{m} \log(\frac{1}{\delta})$ for every $i \in [k]$, where $\delta := \frac{1}{8d\gamma_1^2} \times \frac{\gamma_1^2 - \gamma_d^2}{(\sigma_1 - \sigma_d)^2}$.*

We now state our main result.

**Theorem 2.2** **(Main result).** *Let $T > 0$. Given a rectangular matrix $A \in \mathbb{R}^{m \times d}$ with singular values $\sigma_1 \geq ... \geq \sigma_d \geq 0$ and corresponding orthonormal right-singular vectors $v_1, ... v_d$ (and denote $V := [v_1, ... v_d] \in \mathbb{R}^{d \times d}$). Let $G$ be a matrix with i.i.d. $N(0, 1)$ entries, and consider the perturbed matrix $\hat{A} := A + \sqrt{T}G \in \mathbb{R}^{m \times d}$.*

*Define $\hat{\sigma}_1 \geq ... \geq \hat{\sigma}_d \geq 0$ to be the singular values of $\hat{A}$ with corresponding orthonormal right-singular vectors $\hat{v}_1, ... \hat{v}_d$ (and denote $\hat{V} := [\hat{v}_1, ... \hat{v}_d]$).*

*Let $\gamma_1 \geq ... \geq \gamma_d \geq 0$ and $k \in [d]$ be any numbers such that $\gamma_i = 0$ for $i > k$, and define $\Gamma := \mathrm{diag}(\gamma_1, ..., \gamma_d)$. Then if $A$ satisfies Assumption 2.1 for $(A, k, T, \sigma, \gamma)$, we have*

$$\mathbb{E}\left[\|\hat{V}\Gamma^\top \Gamma \hat{V}^\top - V\Gamma^\top \Gamma V^\top\|_F^2\right] \leq O\left(\sum_{i=1}^{k} \sum_{j=i+1}^{d} \frac{(\gamma_i^2 - \gamma_j^2)^2}{(\sigma_i - \sigma_j)^2}\right) T. \tag{4}$$

We give an overview of the proof of Theorem 2.2 in Section 4. The full proof is given in Appendix A.

## 2.1 APPLICATION TO SINGULAR SUBSPACE RECOVERY.

To obtain a pertubation bound for the subspace recovery problem, we plug in $\gamma_i = 1$ for all $i \leq k$, and $\gamma_i = 0$ for all $i > k$, into Theorem 2.2.

**Corollary 2.3** **(Subspace recovery).** *Let $T > 0$. Given a rectangular matrix $A \in \mathbb{R}^{m \times d}$ with singular values $\sigma_1 \geq ... \geq \sigma_d \geq 0$ and corresponding right-singular vectors $v_1, ... v_d$. Let $G$ be a matrix with i.i.d. $N(0, 1)$ entries, and consider the perturbed matrix $\hat{A} = A + \sqrt{T}G$.*

*For any $k \in [d]$, define the $d \times k$ matrices $V_k = [v_1, ... v_k]$ and $\hat{V}_k = [\hat{v}_1, ... \hat{v}_k]$ where $\hat{v}_1, \cdots, \hat{v}_k$ denote the right-singular vectors of $\hat{A}$ corresponding to its top-$k$ singular values. Then if $A$ satisfies Assumption 2.1$(A, k, T, \sigma, \gamma)$ where $\gamma = (1, \cdots, 1, 0, \cdots, 0)$ is the vector with the first $k$ entries equal to 1, we have*

$$\mathbb{E}\left[\|\hat{V}_k \hat{V}_k^\top - V_k V_k^\top\|_F\right] \leq O\left(\frac{\sqrt{kd}}{\sigma_k - \sigma_{k+1}}\sqrt{T}\right). \tag{5}$$

*Moreover, if we further have that $\sigma_i - \sigma_{i+1} \geq \Omega(\sigma_k - \sigma_{k+1})$ for all $i \leq k$, then*

$$\mathbb{E}\left[\|\hat{V}_k \hat{V}_k^\top - V_k V_k^\top\|_F\right] \leq O\left(\frac{\sqrt{d}}{\sigma_k - \sigma_{k+1}}\sqrt{T}\right). \tag{6}$$

The proof of Corollary 2.3 is given in Appendix B. Corollary 2.3 improves, in the setting where the perturbation $G$ is a Gaussian random matrix, by a factor of $\frac{\sqrt{m}}{\sqrt{d}}$ (in expectation) on the bound $\|\hat{V}_k \hat{V}_k^\top - V_k V_k^\top\|_F \leq O(\frac{\sqrt{km}}{\sigma_k - \sigma_{k+1}}\sqrt{T})$ w.h.p. implied by the Davis-Kahan-Wedin sine-Theta theorem Davis & Kahan (1970); Wedin (1972), whenever Assumption 2.1 is satisfied. If we also have that $\sigma_i - \sigma_{i+1} \geq \Omega(\sigma_k - \sigma_{k+1})$ for all $i \leq k$ (as is the case for many real-world datasets which may exhibit exponential decay in their singular values), the improvement is $\sqrt{k}\frac{\sqrt{m}}{\sqrt{d}}$.

Moreover, Corollary 2.3 also improves, in the setting where the perturbation $G$ is a Gaussian random matrix, by a factor of $\sqrt{k}\frac{\sqrt{m}}{\sqrt{d}}$, on the bound of $\|\hat{V}_k\hat{V}_k^\top - V_kV_k^\top\|_F \leq O(\frac{\sqrt{mk}}{\sigma_k}\sqrt{T})$ w.h.p. implied by Theorem 18 of O'Rourke et al. (2018), and by a factor of $\frac{\sqrt{m}}{\sqrt{d}}$, on the bound of $\|\hat{V}_k\hat{V}_k^\top - V_kV_k^\top\|_F \leq O(\frac{\sqrt{m}\sqrt{k}}{\sigma_k}\sqrt{T})$ w.h.p. implied by Theorem 7 of O'Rourke et al. (2023), when Assumption 2.1 is satisfied and e.g. $\sigma_k - \sigma_{k+1} = \Omega(\sigma_k)$ (as is also the case for many real-world datasets). This can lead to a large improvement in many applications, as one oftentimes has that the number $m$ of rows in the data matrix (corresponding to the number of datapoints) is much larger than the number of columns $d$ (which oftentimes correspond to different features in the data).

Finally, Corollary 2.3 also implies the same upper bound on the expected spectral norm, since $\mathbb{E}[\|V_kV_k^\top - \hat{V}_k\hat{V}_k^\top\|_2] \leq \mathbb{E}[\|V_kV_k^\top - \hat{V}_k\hat{V}_k^\top\|_F]$. Thus it improves, e.g., by a factor of $\frac{\sqrt{m}}{\sqrt{d}\sqrt{k}}$ (in expectation) on the spectral norm bound $\|\hat{V}_k\hat{V}_k^\top - V_kV_k^\top\|_2 \leq O(\frac{\sqrt{m}\sqrt{k}}{\sigma_k}\sqrt{T})$ w.h.p. implied by Theorem 7 of O'Rourke et al. (2023), whenever Assumption 2.1 is satisfied, $\sigma_k - \sigma_{k+1} = \Omega(\sigma_k)$ and $m > dk$.

## 2.2 APPLICATION TO RANK-$k$ COVARIANCE MATRIX APPROXIMATION.

To obtain a perturbation bound for the rank-$k$ covariance matrix approximation problem, we plug in $\gamma_i = \sigma_i$ for all $i \leq k$, and $\gamma_i = 0$ for all $i > k$, into Theorem 2.2.

**Corollary 2.4** (**Rank-$k$ covariance matrix approximation**). *Let $T > 0$. Given a rectangular matrix $A \in \mathbb{R}^{m \times d}$ with singular values $\sigma_1 \geq ... \geq \sigma_d \geq 0$ and with right-singular vectors $v_1, ...v_d$, where we define $V := [v_1, ...v_d] \in \mathbb{R}^{d \times d}$. Let $G$ be a matrix with i.i.d. $N(0,1)$ entries, and consider the perturbed matrix that outputs $\hat{A} = A + \sqrt{T}G$.*

*For any $k \in [d]$, define $\Sigma_k := diag(\sigma_1, ..., \sigma_k, 0, ...0)$. Define $\hat{\sigma}_1 \geq ... \geq \hat{\sigma}_d \geq 0$ to be the singular values of $\hat{A}$ with corresponding orthonormal right-singular vectors $\hat{v}_1, ...\hat{v}_d$, where we define $\hat{V} := [\hat{v}_1, ...\hat{v}_d]$, and define $\hat{\Sigma}_k := diag(\hat{\sigma}_1, ..., \hat{\sigma}_k, 0, ...0)$. Then if $A$ satisfies Assumption 2.1 for $(A, k, T, \sigma, \gamma)$ for $\gamma = (\sigma_1, \cdots, \sigma_k, 0, \cdots, 0)$, we have*

$$\mathbb{E}\left[\|\hat{V}\hat{\Sigma}_k^\top\hat{\Sigma}_k\hat{V}^\top - V\Sigma_k^\top\Sigma_kV^\top\|_F^2\right] \leq O\left(d\|\Sigma_k\|_F^2 + k\sum_{j=k+1}^{d}\left(\sigma_k\frac{\sigma_k}{\sigma_k - \sigma_j}\right)^2\right)T. \quad (7)$$

The proof of Corollary 2.4 is given in Appendix C. In particular, Corollary 2.4 implies that

$$\sqrt{\mathbb{E}\left[\|\hat{V}\hat{\Sigma}_k^\top\hat{\Sigma}_k\hat{V}^\top - V\Sigma_k^\top\Sigma_kV^\top\|_F^2\right]} \leq O\left(\sqrt{k}\sqrt{d}\left(\sigma_1 + \sigma_k\frac{\sigma_k}{\sigma_k - \sigma_{k+1}}\right)\right)\sqrt{T}.$$

Corollary 2.4 improves, in the setting where the perturbation $G$ is a Gaussian random matrix, by a factor of $\frac{\sqrt{m}}{\sqrt{d}}$ (in expectation) on the bound of $\|\hat{V}\hat{\Sigma}_k^\top\hat{\Sigma}_k\hat{V}^\top - V\Sigma_k^\top\Sigma_kV^\top\|_F \leq O(k^{1.5}\sqrt{m}\sqrt{T}\sigma_1 + \sigma_k^2\frac{\sqrt{k}\sqrt{m}}{\sigma_k - \sigma_{k+1}}\sqrt{T})$ w.h.p. implied by the Davis-Kahan-Wedin sine-Theta theorem Davis & Kahan (1970); Wedin (1972) whenever Assumption 2.1 is satisfied (see Appendix D for details). If we also have that $\sigma_k - \sigma_{k+1} = \Omega(\sigma_k)$, the improvement is $\frac{\sqrt{m}}{\sqrt{d}}k$.

Moreover, Corollary 2.4 also improves, when the perturbation is Gaussian, by a factor of $\frac{\sqrt{m}}{\sqrt{d}}\sqrt{k}$ (in expectation) on the bound implied by Theorem 18 of O'Rourke et al. (2018) whenever e.g. $\sigma_k - \sigma_{k+1} = \Omega(\sigma_k)$, as in this setting their bound implies $\|\hat{V}\hat{\Sigma}_k^\top\hat{\Sigma}_k\hat{V}^\top - V\Sigma_k^\top\Sigma_kV^\top\|_F \leq O\left(\sigma_1 k\sqrt{m}\sqrt{T}\right)$ w.h.p. (see Appendix D for details).

**Remark 2.5** (**Tightness in full-rank special case**). *In the special case where $k = d$, we have $\|(A+\sqrt{T}G)^\top(A+\sqrt{T}G) - A^\top A\|_F = \|\sqrt{T}A^\top G + \sqrt{T}G^\top A + TG^\top G\|_F = \Theta(\|A^\top G\|_F\sqrt{T}) = \Theta(\|\Sigma_d\|_F\sqrt{d}\sqrt{T})$ w.h.p. Thus, Corollary 2.4 is tight in this special case. The last equality above holds w.h.p. because $\|A^\top G\|_F^2 = tr(G^\top AA^\top G) = tr(G^\top\Sigma_d\Sigma_d^\top G) = tr(\Sigma_d\Sigma_d^\top GG^\top) = \|\Sigma_d\|_F^2 d$ w.h.p., where we may assume without loss of generality that $A$ is a diagonal matrix because the distribution of $G$ is invariant w.r.t. multiplication by orthogonal matrices.*

## 2.3 APPLICATIONS TO DIFFERENTIAL PRIVACY

In many applications, datasets contain sensitive information. For instance, this may be the case for medical applications where datasets may contain sensitive information about individual patients. In such applications, one can add random noise to the dataset (or, more generally, add random noise to a machine learning algorithm trained on this dataset) to "hide" private information about individuals.

In high-dimensional statistics, one oftentimes cares only about the covariance between a subset $S_1$ of $m$ "input" features and another (possibly, but not necessarily, disjoint) subset $S_2$ of $d$ features (which may correspond to "output" features or labels to be predicted). Differential privacy can be used here to calculate private covariance estimates, especially in settings where the data is too high-dimensional to compute an the full symmetric covariance matrix, as privatizing such a high-dimensional matrix may require adding an unnecessarily large amount of noise. For instance, in Biomedical and Genomics datasets which involve gene expression data, covariances between different features may be stored as a rectangular matrix $A$ where rows represent genes and columns represent disease conditions (see e.g. Patnaik et al. (2012)). Applying DP PCA to these matrices enables privacy-preserving analysis, without exposing sensitive information about individuals.

For any $\varepsilon, \delta > 0$, a randomized mechanism $\mathcal{M}$ is said to be $(\varepsilon, \delta)$-differentially private Dwork (2006) Dwork et al. (2006) if for any two neighboring datasets $D, D' \in \mathcal{D}$ one has $\mathbb{P}(\mathcal{M}(D) \in S) \leq e^\varepsilon \mathbb{P}(\mathcal{M}(D') \in S) + \delta$. Datasets $D, D'$ are said to be neighbors if they differ by at most one datapoint. The *sensitivity* of a function $f : \mathcal{D} \to \mathbb{R}^{m \times d}$ is defined as the supremum of $\|f(D) - f(D')\|_F$ over all neighboring $D, D' \in \mathcal{D}$. Following, e.g., Dwork et al. (2014), we assume the input matrix $A$ is a function of the dataset, $A = f(D)$, where $f$ has sensitivity at most 1 (see also e.g. Kapralov & Talwar (2013); Amin et al. (2019); Mangoubi & Vishnoi (2022); Mangoubi et al. (2022)). To ensure that the sensitivity is $\leq 1$, a standard preprocessing step is to "clip" the datapoints such that each datapoint $x \in D$ has length at most $\|x\| \leq 1$. This ensures that, whenever $A = f(D)$ arises from a 1-Lipschitz function $f$, the sensitivity of this function $f$ will be $\leq 1$. For instance, if $A$ is a rectangular covariance matrix arising from data matrices $X \in \mathbb{R}^{N \times m}$ and $Y \in \mathbb{R}^{N \times d}$ (whose collumns correpond to subsets of size $m$ and $d$ of the features in a dataset, and rows are datapoints), where $A = X^\top Y$, then $A$ is a function $f((X, Y)) = X^\top Y$ which is 1-Lipschitz in each datapoint.

One of the most popular methods of privatizing a dataset is the Gaussian mechanism, a randomized mechanism which adds iid Gaussian noise to each entry of the data matrix Dwork et al. (2006). Prior works (e.g., Dwork et al. (2014); Mangoubi & Vishnoi (2022; 2023; In Press)) have provided utility bounds for a version of the Gaussian mechanism in the special case when $A$ is a *symmetric* matrix, and when the noise $G$ added to this matrix is a *symmetric* Gaussian random matrix. However, in many applications including those mentioned above, it is oftentimes desirable to output a privatized version of a *rectangular* matrix $A \in \mathbb{R}^{m \times d}$.

The Gaussian mechanism adds Gaussian noise $A + \sqrt{T}G$ to the output of $f(D) = A$, where each entry of the random matrix $G \in \mathbb{R}^{m \times d}$ is i.i.d. $N(0, 1)$, for some $T > 0$. If $f$ has sensitivity at most 1 (as is the case in the above examples), and one sets $T = \frac{2 \log(\frac{1.25}{\delta})}{\varepsilon^2}$ then the Gaussian mechanism can be shown to satisfy $(\varepsilon, \delta)$-differential privacy Dwork et al. (2006). Our bounds in Corollary 2.3 therefore immediately imply a bound on the Frobenius-norm utility of the subspace spanned by the top-$k$ right-singular vectors of the output $A + \sqrt{T}G$ of the Gaussian mechanism, when the Gaussian mechanism is applied to a rectangular matrix $A \in \mathbb{R}^{m \times d}$. In particular, Corollary 2.3 implies a utility bound of $\mathbb{E}\left[\|\hat{V}_k \hat{V}_k^\top - V_k V_k^\top\|_F\right] \leq O\left(\frac{\sqrt{d}}{\sigma_k - \sigma_{k+1}} \sqrt{T}\right) = O\left(\frac{\sqrt{d}}{\sigma_k - \sigma_{k+1}} \frac{\sqrt{2 \log(\frac{1.25}{\delta})}}{\varepsilon}\right)$ for the Gaussian mechanism with $(\varepsilon, \delta)$-differential privacy, whenever the singular values $\sigma_1 \geq \cdots \geq \sigma_d$ of $A$ satisfy Assumption 2.1 and, e.g. $\sigma_i - \sigma_{i+1} \geq \Omega(\sigma_k - \sigma_{k+1})$ for all $i \leq k$.

This improves by a factor of $\sqrt{m}\sqrt{k}/\sqrt{d}$ (in expectation) on the bound $\|\hat{V}_k \hat{V}_k^\top - V_k V_k^\top\|_F \leq O(\frac{\sqrt{km}}{\sigma_k - \sigma_{k+1}} \frac{\sqrt{2 \log(\frac{1.25}{\delta})}}{\varepsilon})$ implied by the Davis-Kahan-Wedin sine-Theta theorem Davis & Kahan (1970); Wedin (1972), and by a factor of $\sqrt{m}\sqrt{k}/\sqrt{d}$, on the bound of $\|\hat{V}_k \hat{V}_k^\top - V_k V_k^\top\|_F \leq O(\frac{\sqrt{m}\sqrt{k}}{\sigma_k} \sqrt{2 \log(\frac{1.25}{\delta})}/\varepsilon)$ implied by Theorem 7 of O'Rourke et al. (2023).

## 3 PRELIMINARIES

In this section, we present preliminary materials used in the proof of our main result. In particular, we present the aforementioned matrix-valued Brownian motion process $\Phi(t)$ in Section 3.1. Next, we present the stochastic differential equations (SDEs) which govern the evolution of the singular values of right-singular vectors of $\Phi(t)$ in Section 3.2.

### 3.1 DYSON-BESSEL PROCESS

We consider the matrix-valued stochastic motion process, $\Phi(t)$, where, for all $t \geq 0$, the entries of $\Phi(t)$ evolve as independent standard Brownian motions with initial condition $\Phi(0) = A$. In particular, at time $t = T$ we have $\Phi(T) = A + \sqrt{T}G$ where $G$ is an $m \times d$ Gaussian random matrix with iid $N(0,1)$ entries.

Recall that $\sigma_1 \geq \ldots \geq \sigma_d$ denote the singular values of $A$. At every time $t > 0$, we denote (with slight abuse of notation) the singular values of $\Phi(t)$ by $\sigma_1(t) \geq \sigma_2(t) \geq \cdots \geq \sigma_d(t)$. In particular $\sigma_i \equiv \sigma_i(0)$ for all $i \in [d]$, and the singular values $\sigma_1(t), \ldots, \sigma_d(t)$ are distinct at every time $t > 0$ with probability 1 (see e.g. Guionnet & Huang (2021)). The matrix-valued Brownian motion $\Phi(t)$ induces stochastic diffusion processes on the singular values $\sigma_i(t)$ and singular vectors $v_i(t)$, referred to as the Dyson-Bessel process. The dynamics of the singular values $\sigma_i(t)$ of the Dyson-Bessel process are given by the following system of stochastic differential equations (see e.g. Norris et al. (1986) or Theorem 1 in Bru (1989)),

$$d\sigma_i(t) = d\beta_{ii}(t) + \left( \frac{1}{2\sigma_i(t)} \sum_{\{j \in [d]: j \neq i\}} \frac{(\sigma_i(t))^2 + (\sigma_j(t))^2}{(\sigma_i(t))^2 - (\sigma_j(t))^2} + \frac{m-1}{2\sigma_i(t)} \right) dt, \qquad \forall 1 \leq i \leq d, \quad (8)$$

where $\beta_{ii}, 1 \leq i \leq d$ is a family of independent one-dimensional Brownian motions.

### 3.2 RIGHT SINGULAR VECTOR SDE

The dynamics of right-singular vectors $v_i(t)$ of the Dyson-Bessel process are governed by the following stochastic differential equations (see e.g. Norris et al. (1986) or Theorem 2 in Bru (1989)),

$$dv_i(t) = \sum_{\{j \in [d]: j \neq i\}} v_j(t) \sqrt{\frac{(\sigma_j(t))^2 + (\sigma_i(t))^2}{((\sigma_j(t))^2 - (\sigma_i(t))^2)^2}} d\beta_{ji}(t) - \frac{v_i(t)}{2} \frac{(\sigma_j(t))^2 + (\sigma_i(t))^2}{((\sigma_j(t))^2 - (\sigma_i(t))^2)^2} dt$$

$$= \sum_{\{j \in [d]: j \neq i\}} v_j(t) c_{ij}(t) d\beta_{ji}(t) - \frac{v_i(t)}{2} c_{ij}^2(t) dt, \qquad \forall 1 \leq i \leq d, \quad (9)$$

where $\beta_{ij}(t), 1 \leq i < j \leq d$, is a family of independent standard one-dimensional Brownian motions, and the $\beta_{ij}(t)$ form a skew-symmetric matrix, i.e. $\beta_{ij}(t) = -\beta_{ji}(t)$ for all $t \geq 0$. For convenience, in the above equation, we denote $c_{ij}(t) := \sqrt{\frac{(\sigma_j(t))^2 + (\sigma_i(t))^2}{((\sigma_j(t))^2 - (\sigma_i(t))^2)^2}} = c_{ji}(t)$ for all $i, j \in [d]$.

### 3.3 ITO'S LEMMA

We will also use the following result from stochastic Calculus, Ito's Lemma, which is a generalization of the chain rule in deterministic calculus.

**Lemma 3.1 (Ito's Lemma Itô (1951)).** *Let $f : \mathbb{R}^d \to \mathbb{R}$ be a second-order differentiable function, and let $X(t)$ be a diffusion process on $\mathbb{R}^d$. Then*

$$df(X_t) = (\nabla f(X_t))^\top dX_t + \tfrac{1}{2}(dX_t)^\top (\nabla^2 f(X_t)) dX_t \qquad \forall t \geq 0.$$

### 3.4 OTHER PRELIMINARIES

We will use the following deterministic eigenvalue perturbation bound

**Lemma 3.2 (Weyl's Inequality Weyl (1912)).** *Let $A, E \in \mathbb{R}^{m \times d}$ is a matrix. Denote by $\sigma_1 \geq \ldots \geq \sigma_d$ the singular values of $A$ and by $\hat{\sigma}_1 \geq \ldots \geq \hat{\sigma}_d$ the singular values of $A + E$. Then*

$$|\sigma_i - \hat{\sigma}_i| \leq \|E\|_2 \qquad \forall i \in [d].$$

The following concentration bound, Theorem 4.4.5 of Vershynin (2018) applied to Gaussian random matrices, will allow us to bound the spectral norm of the Gaussian perturbation $G$ (which in turn will allow us to apply equation 3.2 to bound the perturbations to eigenvalues).

**Lemma 3.3** (**Spectral-norm concentration bound for Gaussian matrices Vershynin (2018)**)**.** *If $G \in \mathbb{R}^{m \times d}$ is a Gaussian random matrix with iid $N(0,1)$ entries, then $\mathbb{P}(\|G\|_2 > \sqrt{m} + \sqrt{d} + s) < 2e^{-s^2}$ for all $s > 0$.*

# 4 OVERVIEW OF PROOF OF THEOREM 2.2

We present an overview of the proof of Theorem 2.2 along with the main technical lemmas used in the proof. In Steps 1 and 2 we express the perturbed matrix, and its quantities of interest derived from its right-singular vectors, as matrix-valued diffusions. Steps 3, 4, and 5 present the main technical lemmas, and we complete the proof in Step 6. The full proof is given in Appendix A.5.

## 4.1 VIEWING THE PERTURBED MATRIX AS A MATRIX-VALUED BROWNIAN MOTION.

To obtain our bounds, we begin by defining the matrix-valued Brownian motion, $\Phi(t) := A + B(t)$ for all $t \geq 0$, where the entries of $B(t)$ evolve as independent standard Brownian motions initialized at 0. In particular, at time $t = 0$ we have $\Phi(0) = A$, and at time $t = T$ we have $\Phi(T) = A + \sqrt{T}G$ where $G$ is an $m \times d$ Gaussian random matrix with iid $N(0,1)$ entries.

## 4.2 PROJECTING THE MATRIX BROWNIAN MOTION ONTO THE ORTHOGONAL ORBIT $\mathcal{O}_{\Gamma^2}$.

Denote by $A = U\Sigma V^\top$ and $\hat{A} = \hat{U}\hat{\Sigma}\hat{V}^\top$ singular value decompositions of $A$ and $\hat{A}$, respectively, where $U, \hat{U} \in \mathrm{O}(m)$, $V, \hat{V} \in \mathrm{O}(d)$, and $\Sigma, \hat{\Sigma} \in \mathbb{R}^{m \times d}$ are diagonal.

Recall that our goal is to bound the quantity $\mathbb{E}[\|\hat{V}\Gamma^\top\Gamma\hat{V}^\top - V\Gamma^\top\Gamma V^\top\|_F]$, where $A^\top A = V\Sigma^\top\Sigma V^\top$ and $\hat{A}^\top\hat{A} = \hat{V}\hat{\Sigma}^\top\hat{\Sigma}\hat{V}^\top$ are eigenvalue decompositions of $A^\top A$ and $\hat{A}^\top\hat{A}$. To obtain a bound on this quantity, we first define a stochastic process $\Psi(t)$ for which $\Psi(0) = V\Gamma^\top\Gamma V^\top$ and $\Psi(T) = \hat{V}\Gamma^\top\Gamma\hat{V}^\top$. We then bound the expected Frobenius distance

$$\mathbb{E}[\|\hat{V}\Gamma^\top\Gamma\hat{V}^\top - V\Gamma^\top\Gamma V^\top\|_F] = \mathbb{E}[\|\Psi(T) - \Psi(0)\|_F]$$

by integrating the stochastic derivative of $\Psi(t)$ over the time period $[0, T]$.

Towards this, at every time $t \geq 0$, define $\Phi(t) := U(t)\Sigma(t)V(t)^\top$ to be a singular value decomposition of the rectangular matrix $\Phi(t)$, where $\Sigma(t) \in \mathbb{R}^{m \times d}$ is a diagonal matrix whose diagonal entries are the singular values $\sigma_1(t) \geq \cdots \geq \sigma_d(t)$ of $\Phi(t)$. $V(t) = [v_1(t), \cdots, v_d(t)]$ is a $d \times d$ orthogonal matrix whose columns $v_1(t), \cdots, v_d(t)$ are the corresponding right-singular vectors of $\Phi(t)$. $V(t) \in \mathrm{O}(m)$ is an $m \times m$ orthogonal matrix whose columns are left-singular vectors of $\Phi(t)$.

At every time, denote by $\Psi(t) \in \mathcal{O}_{\Gamma^2}$ to be the symmetric matrix with given eigenvalues $\Gamma^\top\Gamma$ and eigenvectors given by the columns of $V(t)$:

$$\Psi(t) := V(t)\Gamma^\top\Gamma V(t)^\top, \forall t \in [0, T].$$

In other words, $\Psi(t) \in \mathcal{O}_{\Gamma^2}$ is the Frobenius-distance minimizing projection of the matrix Brownian motion $\Phi(t)$ onto the orthogonal orbit manifold $\mathcal{O}_{\Gamma^2}$.

## 4.3 DERIVING AN EXPRESSION FOR THE STOCHASTIC DERIVATIVE $\mathrm{d}\Psi(t)$.

To bound the expected squared Frobenius distance $\mathbb{E}\left[\|\Psi(T) - \Psi(0)\|_F^2\right]$ we would like to express it as an integral in terms of the stochastic derivative of $\Phi(t)$.

Towards this end, we use the stochastic differential equations which govern the evolution of the eigenvectors of the Dyson-Bessel process equation 9 to derive an expression for the stochastic

derivative $\mathrm{d}\Psi(t)$ of the matrix diffusion $\Psi(t)$ (Lemma A.2),

$$
\begin{aligned}
\mathrm{d}\Psi(t) &= \sum_{i=1}^{d} \gamma_i^2 \mathrm{d}(v_i(t)v_i^\top(t)) \\
&= \sum_{i=1}^{d}\sum_{j\neq i}(\gamma_i^2 - \gamma_j^2)\left[\frac{c_{ij}(t)}{2}\mathrm{d}\beta_{ji}(t)(v_i(t)v_j^\top(t) + v_j(t)v_i^\top(t)) - c_{ij}^2(t)\mathrm{d}t(v_i(t)v_i^\top(t))\right].
\end{aligned} \tag{10}
$$

### 4.4 BOUNDING THE SINGULAR VALUE GAPS.

The above equation (equation 10) for the stochastic derivative $\mathrm{d}\Psi(t)$ includes terms $c_{ij}(t)$, whose magnitude is proportional to the inverse of the gaps in the squared singular values $\sigma_i^2(t) - \sigma_j^2(t)$ for each $i, j \in [d]$. In order to bound these terms, we use Weyl's inequality equation 3.2 together with standard concentration bounds for the spectral norm of Gaussian random matrices (Lemma 3.3), to show that the gaps $\sigma_i(t) - \sigma_j(t)$ in the top $k + 1$ singular values satisfy (Lemma A.3),

$$
\sigma_i(t) - \sigma_j(t) \geq \frac{1}{2}(\sigma_i - \sigma_j) \qquad \forall t \in [0, T],\ i < j \leq k + 1
$$

with high probability at least $1 - \delta$, provided that the initial gaps are sufficiently large to satisfy Assumption 2.1($A, k, T, \sigma, \gamma$). This implies that, with high probability at least $1 - \delta$, the inverse-eigenvalue gap terms in equation 10 satisfy (Lemma A.4)

$$
c_{ij}(t) = \sqrt{\frac{(\sigma_j(t))^2 + (\sigma_i(t))^2}{((\sigma_j(t))^2 - (\sigma_i(t))^2)^2}} \leq \frac{4}{\sigma_i - \sigma_j}, \qquad \forall i < j,\ t \in [0, T]. \tag{11}
$$

### 4.5 INTEGRATING THE STOCHASTIC DERIVATIVE OF $\mathrm{D}\Psi(t)$ OVER THE TIME INTERVAL $[0, T]$.

Next we express the expected squared Frobenius distance $\mathbb{E}\left[\|\Psi(T) - \Psi(0)\|_F^2\right]$ as an integral $\mathbb{E}\left[\|\Psi(T) - \Psi(0)\|_F^2\right] = \mathbb{E}\left[\|\int_0^T \mathrm{d}\Psi(t)\|_F^2\right]$.

Next, we apply Ito's Lemma (Lemma 3.1) to $f(\Psi(t))$ where $f(X) := \|\cdot\|_F^2$, and plug in our high-probability bound on the inverse eigenvalue gap terms $c_{ij}(t)$ equation 11, to derive an upper bound for the integral $\mathbb{E}\left[\|\int_0^T \mathrm{d}\Psi(t)\|_F^2\right]$, which gives roughly (Lemma A.5)

$$
\begin{aligned}
&\mathbb{E}\left[\|\Psi(T) - \Psi(0)\|_F^2\right] \\
&\leq 32\int_0^T \mathbb{E}\left[\sum_{i=1}^{d}\sum_{j\neq i}(\gamma_i^2 - \gamma_j^2)^2 c_{ij}^2(t)\right]\mathrm{d}t + 32T\int_0^T \mathbb{E}\left[\sum_{i=1}^{d}\left(\sum_{j\neq i}(\gamma_i^2 - \gamma_j^2)^2 c_{ij}^2(t)\right)^2\right]\mathrm{d}t \\
&\leq 32\int_0^T \mathbb{E}\left[\sum_{i=1}^{d}\sum_{j\neq i}\frac{(\gamma_i^2 - \gamma_j^2)^2}{(\sigma_i - \sigma_j)^2}\right]\mathrm{d}t + 32T\int_0^T \mathbb{E}\left[\sum_{i=1}^{d}\left(\sum_{j\neq i}\frac{|\gamma_i^2 - \gamma_j^2|}{(\sigma_i - \sigma_j)^2}\right)^2\right]\mathrm{d}t.
\end{aligned} \tag{12}
$$

Noting that the second term on the right-hand side of (12) is at least as small as the first term, and applying the Cauchy-Schwarz inequality to the second term, we get that (Theorem 2.2),

$$
\mathbb{E}\left[\|\hat{V}\Gamma^\top\Gamma\hat{V}^\top - V\Gamma^\top\Gamma V^\top\|_F^2\right] = \mathbb{E}\left[\|\Psi(T) - \Psi(0)\|_F^2\right] \leq O\left(\sum_{i=1}^{k}\sum_{j=i+1}^{d}\frac{(\gamma_i^2 - \gamma_j^2)^2}{(\sigma_i - \sigma_j)^2}\right)T.
$$

## 5 CONCLUSION

In this paper, we obtain Frobenius-norm bounds on the perturbation to the singular subspace spanned by the top-$k$ singular vectors of a matrix $A \in \mathbb{R}^{m \times d}$, when $A$ is perturbed by an $m \times d$ Gaussian random matrix. Our bounds improve, in many settings where the perturbation is Gaussian, on bounds

implied by previous works, by a factor of roughly $\frac{\sqrt{m}}{\sqrt{d}}$. This may lead to a large improvement in many applications, as one oftentimes has that the number $m$ of rows in the data matrix (corresponding to the number of datapoints) is much larger than the number of columns $d$ (which oftentimes correspond to different features in the data). To obtain our bounds we view use tools from stochastic calculus to track the evolution of the subspace spanned by the top-$k$ singular vectors.

On the other hand, we note that our bounds assume that the top-$k$ singular value gaps of $A$ are roughly $\Omega(\sqrt{m})$; while this assumption may hold in settings where the data matrix has fast-decaying singular values, it would be interesting to see if it is possible to relax this assumption. Moreover, we note that our bounds only apply in the special case when the perturbation $G$ is Gaussian, and it would be interesting to see whether our bounds can be extended to other random matrix distributions.

ACKNOWLEDGMENTS

This work was supported in part by OM's Google Research Scholar award.

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

CONTENTS

# A   PROOF THEOREM 2.2

## A.1   PROOF OF LEMMA A.1

We first decomposite the matrix $\Psi(t)$ as a sum of its right-singular vectors: $\Psi(t) = \sum_{i=1}^{d} \gamma_i^2 (v_i(t) v_i^\top(t))$. Thus we have

$$\mathrm{d}\Psi(t) = \sum_{i=1}^{d} \gamma_i^2 \mathrm{d}(v_i(t) v_i^\top(t)) \tag{13}$$

We begin by computing the stochastic derivative $\mathrm{d}v_i(t) v_i^\top(t)$ for each $i \in [d]$, by applying the formula in (9), together with Ito's Lemma (Lemma 3.1).

**Lemma A.1** (Stochastic derivative of $v_i(t) v_i(t)^\top$). *For all $t \in [0, T]$,*

$$\mathrm{d}\left(v_i(t) v_i^\top(t)\right) = \sum_{j \neq i} v_j(t) c_{ij}(t) \mathrm{d}\beta_{ji}(t) - \frac{1}{2} v_i(t) \sum_{j \neq i} c_{ij}^2(t) \mathrm{d}t.$$

*Proof.* The dynamic of right-singular vectors Bru (1989) are the following:

$$\mathrm{d}v_i(t) = \sum_{j \neq i} v_j(t) \sqrt{\frac{\lambda_j(t) + \lambda_i(t)}{(\lambda_j(t) - \lambda_i(t))^2}} \mathrm{d}\beta_{ji}(t) - \frac{1}{2} v_i(t) \sum_{j \neq i} \frac{\lambda_j(t) + \lambda_i(t)}{(\lambda_j(t) - \lambda_i(t))^2} \mathrm{d}t$$

$$= \sum_{j \neq i} v_j(t) c_{ij}(t) \mathrm{d}\beta_{ji}(t) - \frac{1}{2} v_i(t) \sum_{j \neq i} c_{ij}^2(t) \mathrm{d}t.$$

Thus, we have

$$\mathrm{d}\left(v_i(t) v_i^\top(t)\right) = (v_i(t) + \mathrm{d}v_i(t))(v_i(t) + \mathrm{d}v_i(t))^\top - v_i(t) v_i^\top(t)$$

$$= \left(v_i(t) + \sum_{j \neq i} v_j(t) c_{ij}(t) \mathrm{d}\beta_{ji}(t) - \frac{1}{2} v_i(t) \sum_{j \neq i} c_{ij}^2 \mathrm{d}t\right)$$

$$\times \left(v_i(t)^\top + \sum_{j \neq i} v_j(t)^\top c_{ij}(t) \mathrm{d}\beta_{ji}(t) - \frac{1}{2} v_i(t)^\top \sum_{j \neq i} c_{ij}^2(t) \mathrm{d}t\right) - v_i(t) v_i(t)^\top$$

$$= v_i(t) \left(\sum_{j \neq i} v_j^\top(t) c_{ij}(t) \mathrm{d}\beta_{ji}(t)\right) - \frac{1}{2} v_i(t) v_i^\top(t) \sum_{j \neq i} c_{ij}^2(t) \mathrm{d}t + \left(\sum_{j \neq i} v_j(t) c_{ij}(t) \mathrm{d}\beta_{ji}(t)\right) v_i^\top(t)$$

$$+ \left(\sum_{j \neq i} v_j(t) c_{ij}(t) \mathrm{d}\beta_{ji}(t)\right) \left(\sum_{j \neq i} v_j^\top(t) c_{ij}(t) \mathrm{d}\beta_{ji}(t)\right) - \frac{1}{2} v_i(t) v_i^\top(t) \sum_{j \neq i} c_{ij}^2(t) \mathrm{d}t + o(\mathrm{d}t)$$

$$= v_i(t) \left(\sum_{j \neq i} v_j^\top(t) c_{ij}(t) \mathrm{d}\beta_{ji}(t)\right) + \left(\sum_{j \neq i} v_j(t) c_{ij}(t) \mathrm{d}\beta_{ji}(t)\right) v_i^\top(t) - v_i(t) v_i^\top(t) \sum_{j \neq i} c_{ij}^2(t) \mathrm{d}t$$

$$+ \sum_{k \neq i} \sum_{j \neq i} v_k(t) v_j^\top(t) c_{ik}(t) c_{ij}(t) \mathrm{d}\beta_{ki}(t) \mathrm{d}\beta_{ji}(t)$$

$$= v_i(t) \left(\sum_{j \neq i} v_j^\top(t) c_{ij}(t) \mathrm{d}\beta_{ji}(t)\right) + \left(\sum_{j \neq i} v_j(t) c_{ij}(t) \mathrm{d}\beta_{ji}(t)\right) v_i^\top(t) - v_i(t) v_i^\top(t) \sum_{j \neq i} c_{ij}^2(t) \mathrm{d}t$$

$$+ \sum_{k \neq i} \sum_{j \neq i} v_k(t) v_j^\top(t) c_{ik}(t) c_{ij}(t) \mathbb{1}_{\{(kj) = (ii)\}} \mathrm{d}t$$

$$= \sum_{j \neq i} c_{ij}(t) \mathrm{d}\beta_{ji}(t)(v_i(t) v_j^\top(t) + v_j(t) v_i^\top(t)) - \sum_{j \neq i} c_{ij}^2(t) \mathrm{d}t(v_i(t) v_i^\top(t) - v_j(t) v_j^\top(t)).$$

$\square$

## A.2 PROOF OF LEMMA A.2

Recall that

$$\Psi(t) = \sum_{i=1}^{d} \gamma_i^2 (v_i(t)v_i^\top(t)).$$

We now apply Lemma A.1 to compute the stochastic derivative of $\Psi(t)$.

**Lemma A.2** (Stochastic derivative of $\Psi(t)$). *For all $t \in [0, T]$, we have that*

$$\mathrm{d}\Psi(t) = \sum_{i=1}^{d} \sum_{j \neq i} \frac{\gamma_i^2 - \gamma_j^2}{2} \left[ c_{ij}(t)\mathrm{d}\beta_{ji}(t)(v_i(t)v_j^\top(t) + v_j(t)v_i^\top(t)) - c_{ij}^2(t)\mathrm{d}t(v_i(t)v_i^\top(t) - v_j(t)v_j(t)^\top) \right].$$

*Proof.*

$$\mathrm{d}\Psi(t) = \sum_{i=1}^{d} \gamma_i^2 \mathrm{d}(v_i(t)v_i^\top(t))$$

$$= \sum_{i=1}^{d} \gamma_i^2 \left( \sum_{j \neq i} c_{ij}(t)\mathrm{d}\beta_{ji}(t)(v_i(t)v_j^\top(t) + v_j(t)v_i^\top(t)) - \sum_{j \neq i} c_{ij}^2(t)\mathrm{d}t(v_i(t)v_i^\top(t) - v_j(t)v_j^\top(t)) \right)$$

$$= \sum_{i=1}^{d} \sum_{j \neq i} \gamma_i^2 c_{ij}(t)\mathrm{d}\beta_{ji}(t)(v_j(t)v_i^\top(t) + v_i(t)v_j^\top(t)) - \sum_{i=1}^{d} \sum_{j \neq i} \gamma_i^2 c_{ij}^2(t)\mathrm{d}t(v_i(t)v_i^\top(t) - v_j(t)v_j^\top(t))$$

$$= \frac{1}{2} \sum_{i=1}^{d} \sum_{j \neq i} (\gamma_i^2 - \gamma_j^2) c_{ij}(t)\mathrm{d}\beta_{ji}(t)(v_i(t)v_j(t)^\top + v_j(t)v_i^\top(t))$$

$$- \frac{1}{2} \sum_{i=1}^{d} \sum_{j \neq i} (\gamma_i^2 - \gamma_j^2) c_{ij}^2(t)\mathrm{d}t(v_i(t)v_i^\top(t) - v_j(t)v_j(t)^\top)$$

$$\tag{14}$$

The last equality in the block of equations 14 holds for the following reason:

Since $\beta_{ij}(t)$ form a skew-symmetric matrix, i.e. $\beta_{ij}(t) = -\beta_{ji}(t)$ for all $t \geq 0$, we have that $\mathrm{d}\beta_{ij}(t) = -\mathrm{d}\beta_{ji}(t)$ for all $t \geq 0$. Thus, we have that for all $j \neq i$,

$$c_{ij}(t)\mathrm{d}\beta_{ij}(t)(v_j(t)v_i(t)^\top + v_i(t)v_j(t)^\top) = -c_{ij}(t)\mathrm{d}\beta_{ji}(t)(v_j(t)v_i(t)^\top + v_i(t)v_j(t)^\top) \tag{15}$$

Thus, combining the pairs of terms in the first double summation on the r.h.s. of equation 14 with index $(i, j) = (a, b)$ and $(i, j) = (b, a)$ for every $b \neq a$, we have by equation 15 that

$$\sum_{i=1}^{d} \sum_{j \neq i} \gamma_i^2 c_{ij}(t)\mathrm{d}\beta_{ji}(t)(v_j(t)v_i^\top(t) + v_i(t)v_j^\top(t))$$

$$\overset{\text{Eq.15}}{=} \frac{1}{2} \sum_{i=1}^{d} \sum_{j \neq i} (\gamma_i^2 - \gamma_j^2) c_{ij}(t)\mathrm{d}\beta_{ji}(t)(v_i(t)v_j(t)^\top + v_j(t)v_i^\top(t)) \tag{16}$$

Moreover, observe that, for every $i \neq j$,

$$(\gamma_i^2 - \gamma_j^2) c_{ij}^2(t)\mathrm{d}t(v_i(t)v_i(t)^\top - v_j(t)v_j(t)^\top) = (\gamma_j^2 - \gamma_i^2) c_{ij}^2(t)\mathrm{d}t(v_j(t)v_j(t)^\top - v_i(t)v_i(t)^\top). \tag{17}$$

Thus, combining the pairs of terms in the second double summation on the r.h.s. of equation 14 with index $(i, j) = (a, b)$ and $(i, j) = (b, a)$ for every $b \neq a$, we have by equation 17 that

$$\sum_{i=1}^{d} \sum_{j \neq i} \gamma_i^2 c_{ij}^2(t)\mathrm{d}t(v_i(t)v_i^\top(t) - v_j(t)v_j^\top(t)) = \frac{1}{2} \sum_{i=1}^{d} \sum_{j \neq i} (\gamma_i^2 - \gamma_j^2) c_{ij}^2(t)\mathrm{d}t(v_i(t)v_i^\top(t) - v_j(t)v_j(t)^\top)$$

$$\tag{18}$$

Pluggining in equation 16 and equation 18 into the second-to-last equality in the block of equations 14, we get that the last equality in the block of equations 14 holds.

Thus, we have

$$
\begin{aligned}
\mathrm{d}\Psi(t) \stackrel{\text{Eq. 14}}{=} & \frac{1}{2}\sum_{i=1}^{d}\sum_{j\neq i}(\gamma_i^2 - \gamma_j^2)c_{ij}(t)\mathrm{d}\beta_{ji}(t)(v_i(t)v_j(t)^\top + v_j(t)v_i^\top(t)) \\
& - \frac{1}{2}\sum_{i=1}^{d}\sum_{j\neq i}(\gamma_i^2 - \gamma_j^2)c_{ij}^2(t)\mathrm{d}t(v_i(t)v_i^\top(t) - v_j(t)v_j(t)^\top) \\
= & \sum_{i=1}^{d}\sum_{j\neq i}\frac{\gamma_i^2 - \gamma_j^2}{2}\big[c_{ij}(t)\mathrm{d}\beta_{ji}(t)(v_i(t)v_j^\top(t) + v_j(t)v_i^\top(t)) \\
& \qquad\qquad\qquad - c_{ij}^2(t)\mathrm{d}t(v_i(t)v_i^\top(t) - v_j(t)v_j(t)^\top)\big].
\end{aligned}
$$

$\square$

### A.3 PROOFS OF LEMMAS A.3 AND A.4

Next, we show high-probability bounds on the singular gaps $\sigma_i(t) - \sigma_j(t)$ (Lemma A.3) and coefficients $c_{ij}(t)$ (Lemma A.4).

**Lemma A.3** (Bound on singular gaps:)**.** *Suppose that Assumption 2.1 for $(A, k, T, \sigma, \gamma)$ is satisfied. Then for all $t \in [0, T]$, with probability $1 - \delta$ where $\delta := \frac{1}{8d\gamma_1^2} \times \frac{\gamma_1^2 - \gamma_d^2}{(\sigma_1 - \sigma_d)^2}$, we have $|\sigma_i(t) - \sigma_j(t)| \geq \sqrt{t}\frac{1}{2}(\sigma_i - \sigma_j)$ for any $i < j$.*

*Proof.* With probability at least $1 - \delta$, by Lemma 3.3, we have

$$
\|B(t)\|_2 = \|\sqrt{t}G\|_2 \leq \sqrt{t} \times 2\sqrt{\max\{m, d\}}\log(\frac{1}{\delta}) = \sqrt{t} \times 2\sqrt{m}\log(\frac{1}{\delta}),
$$

where $G$ is a matrix with iid $N(0, 1)$ entries.

Thus, by Weyl's inequality (Lemma 3.2), we have that

$$
|\sigma_i(t) - \sigma_i| \stackrel{\text{Lemma3.2}}{\leq} \|B(t)\|_2 \stackrel{\text{Lemma3.3}}{\leq} \sqrt{t}2\sqrt{m}\log(\frac{1}{\delta}) \tag{19}
$$

for all $i \in [d]$ with probability at least $1 - \delta$.

Therefore, we have that

$$
\begin{aligned}
|\sigma_i(t) - \sigma_j(t)| &\geq \sigma_i - \sigma_j - |\sigma_i(t) - \sigma_i| - |\sigma_j(t) - \sigma_j| \\
&\stackrel{\text{Eq. 19}}{\geq} \sigma_i - \sigma_j - \sqrt{t} \times 4\sqrt{m}\log(\frac{1}{\delta}) \\
&\geq \frac{1}{2}(\sigma_i - \sigma_j)
\end{aligned}
$$

with probability at least $1 - \delta$, for any $i < j$ and any $t \in [0, T]$. $\square$

The following proposition shows that the symmetric coefficients $c_{ij}(t)$ are bounded by the reciprocal of the initial singular value gaps.

**Lemma A.4** (Bound of coefficients $c_{ij}(t)$)**.** *Suppose that Assumption 2.1 for $(A, k, T, \sigma, \gamma)$ is satisfied. Then for all $t \in [0, T]$, with probability $1 - \delta$ where $\delta := \frac{1}{8d\gamma_1^2} \times \frac{\gamma_1^2 - \gamma_d^2}{(\sigma_1 - \sigma_d)^2}$, we have*

$$
c_{ij}(t) \leq \frac{4}{\sigma_i - \sigma_j}, \quad \text{for any } i < j.
$$

*Proof.* By Lemma A.3, we have we have with probability at least $1 - \delta$

$$
\begin{aligned}
c_{ij}(t) &= \frac{\sqrt{\sigma_j^2(t) + \sigma_i^2(t)}}{|\sigma_j^2(t) - \sigma_i^2(t)|} \\
&\leq 2 \frac{\sigma_j(t) + \sigma_i(t)}{|\sigma_j(t) - \sigma_i(t)|(\sigma_i(t) + \sigma_i(t))} \\
&= \frac{2}{|\sigma_j(t) - \sigma_i(t)|} = \frac{2}{|\sigma_i(t) - \sigma_j(t)|} \leq \frac{4}{\sigma_i - \sigma_j}, \quad \text{for any } i < j.
\end{aligned}
$$

$\square$

## A.4 PROOF OF LEMMA A.5

Next, to bound the quantity $\mathbb{E}\left[\|\Psi(T) - \Psi(0)\|_F^2\right]$, use Lemma A.2 together with Ito's Lemma (Lemma 3.1), and then apply Lemma A.4 to the resulting expression (Lemma A.5).

**Lemma A.5** (Bound the Frobenius error as an integral of $\Psi(t)$)**.**

$$
\mathbb{E}\left[\|\Psi(T) - \Psi(0)\|_F^2\right]
$$

$$
\leq 16 \int_0^T \mathbb{E}\left[\sum_{i=1}^d \sum_{j \neq i} \frac{(\gamma_i^2 - \gamma_j^2)^2}{(\sigma_i - \sigma_j)^2}\right] \mathrm{d}t + 32T \int_0^T \mathbb{E}\left[\sum_{i=1}^d \left(\sum_{j \neq i} \frac{|\gamma_i^2 - \gamma_j^2|}{(\sigma_i - \sigma_j)^2}\right)^2\right] \mathrm{d}t. \tag{20}
$$

*Proof.* Let $E$ be the event that $|\sigma_i(t) - \sigma_j(t)| \geq \frac{1}{2}(\sigma_i - \sigma_j)$ for any $i < j$ and any $t \in [0, T]$. By Lemma A.3, we have $\mathbb{P}(E) \geq 1 - \delta$.

By Lemma A.2, we have

$$
\|\Psi(T) - \Psi(0)\|_F^2 = \|\int_o^T \mathrm{d}\Psi(t)\|_F^2
$$

$$
\stackrel{\text{Lemma}A.2}{=} \left\| \frac{1}{2} \int_0^T \sum_{i=1}^d \sum_{j \neq i} (\gamma_i^2 - \gamma_j^2) c_{ij}(t) \mathrm{d}\beta_{ji}(t)(v_i(t)v_j(t)^\top + v_j(t)v_i(t)^\top) \right.
$$

$$
\left. - \frac{1}{2} \int_0^T \sum_{i=1}^d \sum_{j \neq i} (\gamma_i^2 - \gamma_j^2) c_{ij}^2(t) \mathrm{d}t (v_i(t)v_i^\top(t) - v_j(t)v_j(t)^\top) \right\|_F^2
$$

$$
\leq 3 \left\| \frac{1}{2} \int_0^T \sum_{i=1}^d \sum_{j \neq i} (\gamma_i^2 - \gamma_j^2) c_{ij}(t) \mathrm{d}\beta_{ji}(t)(v_i(t)v_j(t)^\top + v_j(t)v_i(t)^\top) \right\|_F^2
$$

$$
+ 3 \left\| \frac{1}{2} \int_0^T \sum_{i=1}^d \sum_{j \neq i} (\gamma_i^2 - \gamma_j^2) c_{ij}^2(t) \mathrm{d}t (v_i(t)v_i^\top(t) - v_j(t)v_j(t)^\top) \right\|_F^2
$$

$$
= 3I_2 + 3I_2 \tag{21}
$$

where the inequality holds by the triangle inequality, and where, for convenience, we define

$$
I_1 := \left\| \frac{1}{2} \int_0^T \sum_{i=1}^d \sum_{j \neq i} (\gamma_i^2 - \gamma_j^2) c_{ij}(t) \mathrm{d}\beta_{ji}(t)(v_i(t)v_j(t)^\top + v_j(t)v_i(t)^\top) \right\|_F^2
$$

and

$$
I_2 := \left\| \frac{1}{2} \int_0^T \sum_{i=1}^d \sum_{j \neq i} (\gamma_i^2 - \gamma_j^2) c_{ij}^2(t) \mathrm{d}t (v_i(t)v_i^\top(t) - v_j(t)v_j(t)^\top) \right\|_F^2.
$$

To evaluate the first integral $I_1$, define

$$X(t) := \int_0^T \sum_{i=1}^d \sum_{j \neq i} (\gamma_i^2 - \gamma_j^2) c_{ij}(t) \mathrm{d}\beta_{ji}(t)(v_i(t)v_j(t)^\top + v_j(t)v_i(t)^\top)$$

for all $t > 0$. Then we have that

$$\mathrm{d}X(t) = \sum_{i=1}^d \sum_{j \neq i} (\gamma_i^2 - \gamma_j^2) c_{ij}(t) \mathrm{d}\beta_{ji}(t)(v_i(t)v_j(t)^\top + v_j(t)v_i(t)^\top) = \sum_{i=1}^d \sum_{j \neq i} R_{ji}(t) \mathrm{d}\beta_{ji}(t)$$

where $R_{ji}(t) := (\gamma_i^2 - \gamma_j^2) \times c_{ij}(t) \times (v_i(t)v_j(t)^\top + v_j(t)v_i(t)^\top)$, so its $[l, r]$ component is

$$\mathrm{d}X(t)[l, r] = \sum_{i=1}^d \sum_{j \neq i} R_{ji}(t)[l, r] \mathrm{d}\beta_{ji}(t).$$

Defining the function $f(X) := \|X\|_F^2 := \sum_{l=1}^d \sum_{r=1}^d X^2[l, r]$ and applying Ito's Lemma (Lemma 3.1), we have

$$\mathrm{d}f(X) = \sum_{l=1}^d \sum_{r=1}^d 2X(t)[l, r] \mathrm{d}X(t)[l, r] + \frac{1}{2} \sum_{l=1}^d \sum_{r=1}^d 2\langle \mathrm{d}X(t)[l, r], \mathrm{d}X(t)[l, r]\rangle$$

$$= \sum_{l=1}^d \sum_{r=1}^d 2X(t)[l, r] \sum_{i=1}^d \sum_{j \neq i} R_{ji}(t)[l, r] \mathrm{d}\beta_{ji}(t) + \sum_{l=1}^d \sum_{r=1}^d \sum_{i=1}^d \sum_{j \neq i} R_{ji}^2(t)[l, r] \mathrm{d}t.$$

Thus,

$$\mathbb{E}(I_1 \times 1_E) = \frac{1}{2} \mathbb{E}\left[(f(X(T)) - f(X(0)) \times 1_E\right] = 0 + \frac{1}{2} \mathbb{E}[\int_0^T \sum_{l=1}^d \sum_{r=1}^d \sum_{i=1}^d \sum_{j \neq i} R_{ji}^2(t)[l, r] \mathrm{d}t \times 1_E]$$

$$= \frac{1}{2} \mathbb{E}[\int_0^T \sum_{l=1}^d \sum_{r=1}^d \sum_{i=1}^d \sum_{j \neq i} \left((\gamma_i^2 - \gamma_j^2) c_{ij}(t)(v_i(t)v_j(t)^\top + v_j(t)v_i(t)^\top)[l, r]\right)^2 \mathrm{d}t \times 1_E]$$

$$= \frac{1}{2} \mathbb{E}[\int_0^T \sum_{i=1}^d \sum_{j \neq i} \sum_{l=1}^d \sum_{r=1}^d \left((\gamma_i^2 - \gamma_j^2) c_{ij}(t)(v_i(t)v_j(t)^\top + v_j(t)v_i(t)^\top)[l, r]\right)^2 \mathrm{d}t \times 1_E]$$

$$= \frac{1}{2} \mathbb{E}[\int_0^T \sum_{i=1}^d \sum_{j \neq i} \|(\gamma_i^2 - \gamma_j^2) c_{ij}(t)(v_i(t)v_j(t)^\top + v_j(t)v_i(t)^\top)\|_F^2 \mathrm{d}t \times 1_E]$$

$$= \frac{1}{2} \int_0^T \mathbb{E}[\sum_{i=1}^d \sum_{j \neq i} (\gamma_i^2 - \gamma_j^2)^2 c_{ij}^2(t) \|(v_i(t)v_j(t)^\top + v_j(t)v_i(t)^\top)\|_F^2 \mathrm{d}t \times 1_E]$$

$$\leq \frac{1}{2} \int_0^T \mathbb{E}[\sum_{i=1}^d \sum_{j \neq i} \frac{16(\gamma_i^2 - \gamma_j^2)^2}{(\sigma_i - \sigma_j)^2} 4 \mathrm{d}t] = 32 \int_0^T \mathbb{E}[\sum_{i=1}^d \sum_{j \neq i} \frac{(\gamma_i^2 - \gamma_j^2)^2}{(\sigma_i - \sigma_j)^2} \mathrm{d}t], \qquad (22)$$

The first inequality holds since the term $\mathbb{E}[\sum_{l=1}^d \sum_{r=1}^d 2X(t)[l, r] \sum_{i=1}^d \sum_{j \neq i} R_{ji}(t)[l, r] \mathrm{d}\beta_{ji}(t)] = 0$ vanishes because $\mathrm{d}\beta_{ji}(t)$ is inedependent of both $X(t)[l, r]$ and $R_{ji}(t)[l, r]$ for every $i, j, l, r$. The last inequality holds since, whenever the event $E$ occurs, we have $|\sigma_i(t) - \sigma_j(t)| \geq \frac{1}{2}(\sigma_i - \sigma_j)$ for any $i < j$ and any $t \in [0, T]$.

For the second integral $I_2$, we have

$$
\begin{aligned}
I_2 &= \left\| \frac{1}{2} \int_0^T \sum_{i=1}^d \sum_{j \neq i} (\gamma_i^2 - \gamma_j^2) c_{ij}^2(t) (v_i(t) v_i^\top(t) - v_j(t) v_j(t)^\top) \mathrm{d}t \right\|_F^2 \\
&= \frac{1}{2} \left\| \int_0^T \sum_{i=1}^d \sum_{j \neq i} (\gamma_i^2 - \gamma_j^2) c_{ij}^2(t) (v_i(t) v_i^\top(t) - v_j(t) v_j(t)^\top) \times 1 \ \mathrm{d}t \right\|_F^2 \\
&\leq \frac{1}{2} \int_0^T \left\| \sum_{i=1}^d \sum_{j \neq i} (\gamma_i^2 - \gamma_j^2) c_{ij}^2(t) (v_i(t) v_i^\top(t) - v_j(t) v_j(t)^\top) \right\|_F^2 \mathrm{d}t \times \int_0^T 1^2 \mathrm{d}t \\
&= \frac{1}{2} T \int_0^T \sum_{i=1}^d \left\| \sum_{j \neq i} (\gamma_i^2 - \gamma_j^2) c_{ij}^2(t) (v_i(t) v_i^\top(t) - v_j(t) v_j(t)^\top) \right\|_F^2 \mathrm{d}t \\
&= \frac{1}{2} T \int_0^T \sum_{i=1}^d \left( \sum_{j \neq i} (\gamma_i^2 - \gamma_j^2) c_{ij}^2(t) \right)^2 \| v_i(t) v_i^\top(t) - v_j(t) v_j(t)^\top \|_F^2 \mathrm{d}t \\
&\leq 2T \int_0^T \sum_{i=1}^d \left( \sum_{j \neq i} (\gamma_i^2 - \gamma_j^2) c_{ij}^2(t) \right)^2 \mathrm{d}t
\end{aligned}
\tag{23}
$$

where the first inequality holds by the Cauchy-Schwartz inequality. The third and fourth equalities hold since $v_i(t) v_i(t)^\top v_j(t) v_j(t)^\top = 0$ for all $i \neq j$. The last equality holds since $\| v_i(t) v_i^\top(t) - v_j(t) v_j(t)^\top \|_F \leq \| v_i(t) v_i^\top(t) \|_F + \| v_j(t) v_j(t)^\top \|_F \leq 2$ because $\| v_i(t) v_i^\top(t) \| = 1$ for all $i \in [d]$.

Whenever the event $E$ occurs we have by the proof of Lemma A.4 that $c_{ij}(t) \leq \dfrac{4}{\sigma_i - \sigma_j}$ for all $i < j$ and all $t \in [0, T]$.

Thus, equation 23 implies that

$$
I_2 \times \mathbb{1}_E \leq 32T \int_0^T \sum_{i=1}^d \left( \sum_{j \neq i} \frac{|\gamma_i^2 - \gamma_j^2|}{(\sigma_i - \sigma_j)^2} \right)^2 \mathrm{d}t.
\tag{24}
$$

We can express $\mathbb{E}[\|\Psi(T) - \Psi(0)\|_F^2]$ as the following sum,

$$
\mathbb{E}[\|\Psi(T) - \Psi(0)\|_F^2] = \mathbb{E}[\|\Psi(T) - \Psi(0)\|_F^2 \times \mathbb{1}_E] + \mathbb{E}[\|\Psi(T) - \Psi(0)\|_F^2 \times \mathbb{1}_{E^c}]
\tag{25}
$$

Combining equation 22 and equation 24, it follows that

$$
\begin{aligned}
\mathbb{E}[\|\Psi(T) - \Psi(0)\|_F^2 \times \mathbb{1}_E] &\leq \mathbb{E}[\|\Psi(T) - \Psi(0)\|_F^2] \\
&\leq \mathbb{E}[I_1 \times \mathbb{1}_E + I_2 \times \mathbb{1}_E] \\
&= \mathbb{E}[I_1 \times \mathbb{1}_E] + \mathbb{E}[I_2 \times \mathbb{1}_E] \\
&\leq 32 \int_0^T \mathbb{E}[\sum_{i=1}^d \sum_{j \neq i} \frac{(\gamma_i^2 - \gamma_j^2)^2}{(\sigma_i - \sigma_j)^2} \mathrm{d}t] + 32T \int_0^T \mathbb{E}[\sum_{i=1}^d \left( \sum_{j \neq i} \frac{|\gamma_i^2 - \gamma_j^2|}{(\sigma_i - \sigma_j)^2} \right)^2] \mathrm{d}t.
\end{aligned}
\tag{26}
$$

Moreover, we have

$$
\begin{aligned}
\mathbb{E}[\|\Psi(T) - \Psi(0)\|_F^2 \times 1_{E^c}] &\leq \mathbb{P}(E^c) \\
&\leq \mathbb{E}[4\|\Psi(T)\|_F^2 + 4\|\Psi(0)\|_F^2 \times 1_{E^c}] \\
&\leq 8d\gamma_1^2 \mathbb{P}(E^c) \\
&\leq 8d\gamma_1^2 \times \delta \\
&\leq \frac{\gamma_1^2 - \gamma_d^2}{(\sigma_1 - \sigma_d)^2} \\
&\leq 32 \int_0^T \mathbb{E}[\sum_{i=1}^d \sum_{j \neq i} \frac{(\gamma_i^2 - \gamma_j^2)^2}{(\sigma_i - \sigma_j)^2} \mathrm{d}t] + 32T \int_0^T \mathbb{E}[\sum_{i=1}^d \left( \sum_{j \neq i} \frac{(\gamma_i^2 - \gamma_j^2)}{(\sigma_i - \sigma_j)^2} \right)^2 ]\mathrm{d}t,
\end{aligned}
\tag{27}
$$

where the fifth inequality holds since $\delta \leq \frac{1}{8d\gamma_1^2} \times \frac{\gamma_1^2 - \gamma_d^2}{(\sigma_1 - \sigma_d)^2}$.

Therefore, plugging equation 26 and equation 27 into equation 25, we have

$$
\mathbb{E}[\|\Psi(T) - \Psi(0)\|_F^2] \leq 32 \int_0^T \mathbb{E}[\sum_{i=1}^d \sum_{j \neq i} \frac{(\gamma_i^2 - \gamma_j^2)^2}{(\sigma_i - \sigma_j)^2} \mathrm{d}t] + 32T \int_0^T \mathbb{E}[\sum_{i=1}^d \left( \sum_{j \neq i} \frac{|\gamma_i^2 - \gamma_j^2|}{(\sigma_i - \sigma_j)^2} \right)^2 ]\mathrm{d}t.
$$

$\square$

## A.5 COMPLETING THE PROOF OF THEOREM 2.2

We now complete the proof of the main result.

*Proof of Theorem 2.2.* From Lemma A.5, we have

$$
\begin{aligned}
\mathbb{E}\left[\|\hat{V}\Gamma^\top \Gamma \hat{V}^\top - V\Gamma^\top \Gamma V^\top\|_F^2\right] &= \mathbb{E}\left[\|\Psi(T) - \Psi(0)\|_F^2\right] \\
&\leq 32 \int_0^T \mathbb{E}[\sum_{i=1}^d \sum_{j \neq i} \frac{(\gamma_i^2 - \gamma_j^2)^2}{(\sigma_i - \sigma_j)^2} \mathrm{d}t] + 32T \int_0^T \mathbb{E}[\sum_{i=1}^d \left( \sum_{j \neq i} \frac{|\gamma_i^2 - \gamma_j^2|}{(\sigma_i - \sigma_j)^2} \right)^2 ]\mathrm{d}t \\
&\leq 64 \int_0^T \mathbb{E}[\sum_{i=1}^d \sum_{j=i+1}^d \frac{(\gamma_i^2 - \gamma_j^2)^2}{(\sigma_i - \sigma_j)^2} \mathrm{d}t] + 64T \int_0^T \mathbb{E}[\sum_{i=1}^d \left( \sum_{j=i+1}^d \frac{|\gamma_i^2 - \gamma_j^2|}{(\sigma_i - \sigma_j)^2} \right)^2 ]\mathrm{d}t \\
&= 64 \int_0^T \mathbb{E}[\sum_{i=1}^k \sum_{j=i+1}^d \frac{(\gamma_i^2 - \gamma_j^2)^2}{(\sigma_i - \sigma_j)^2} \mathrm{d}t] + 64T \int_0^T \mathbb{E}[\sum_{i=1}^k \left( \sum_{j=i+1}^d \frac{|\gamma_i^2 - \gamma_j^2|}{(\sigma_i - \sigma_j)^2} \right)^2 ]\mathrm{d}t \\
&= 64T \sum_{i=1}^k \sum_{j=i+1}^d \frac{(\gamma_i^2 - \gamma_j^2)^2}{(\sigma_i - \sigma_j)^2} + 64T^2 \sum_{i=1}^k \left( \sum_{j=i+1}^d \frac{|\gamma_i^2 - \gamma_j^2|}{(\sigma_i - \sigma_j)^2} \right)^2 \\
&= O\left( \sum_{i=1}^k \sum_{j=i+1}^d \frac{(\gamma_i^2 - \gamma_j^2)^2}{(\sigma_i - \sigma_j)^2} + T \sum_{i=1}^k \left( \sum_{j=i+1}^d \frac{(\gamma_i^2 - \gamma_j^2)}{(\sigma_i - \sigma_j)^2} \right)^2 \right) T.
\end{aligned}
\tag{28}
$$

By the Cauchy-Schwarz inequality, we have that

$$
\begin{aligned}
\left( \sum_{j=i+1}^{d} \frac{(\gamma_i^2 - \gamma_j^2)}{(\sigma_i - \sigma_j)^2} \right)^2 &= \left( \sum_{j=i+1}^{d} \frac{1}{|\sigma_i - \sigma_j|} \times \frac{|\gamma_i^2 - \gamma_j^2|}{|\sigma_i - \sigma_j|} \right)^2 \\
&\leq \left( \sum_{j=i+1}^{d} \frac{1}{(\sigma_i - \sigma_j)^2} \right) \times \left( \sum_{j=i+1}^{d} \frac{(\gamma_i^2 - \gamma_j^2)^2}{(\sigma_i - \sigma_j)^2} \right) \\
&\leq \left( \sum_{j=i+1}^{d} \frac{1}{(\sqrt{d})^2} \right) \times \left( \sum_{j=i+1}^{d} \frac{(\gamma_i^2 - \gamma_j^2)^2}{(\sigma_i - \sigma_j)^2} \right) \\
&\leq \sum_{j=i+1}^{d} \frac{(\gamma_i^2 - \gamma_j^2)^2}{(\sigma_i - \sigma_j)^2}.
\end{aligned}
\tag{29}
$$

Plugging equation 29 into equation 28, we have

$$
\mathbb{E}\left[ \|\hat{V}\Gamma^\top\Gamma\hat{V}^\top - V\Gamma^\top\Gamma V^\top\|_F^2 \right] \leq O\left( \sum_{i=1}^{k} \sum_{j=i+1}^{d} \frac{(\gamma_i^2 - \gamma_j^2)^2}{(\sigma_i - \sigma_j)^2} \right) T.
$$

$\square$

## B  PROOF OF COROLLARY 2.3

*Proof of Corollary 2.3.* To prove Corollary 2.3, we plug in $\gamma_1 = \cdots = \gamma_k = 1$ and $\gamma_{k+1} = \cdots = \gamma_d = 0$ to Theorem 2.2. There are two cases.

In the first case, where $A$ may be any $m \times d$ matrix which satisfies Assumption 2.1, plugging in $\gamma_1 = \cdots = \gamma_k = 1$ and $\gamma_{k+1} = \cdots = \gamma_d = 0$ to Theorem 2.2 we get

$$
\begin{aligned}
\mathbb{E}\left[ \|\hat{V}_k\hat{V}_k^\top - V_kV_k^\top\|_F^2 \right] &= \mathbb{E}\left[ \|\hat{V}\Gamma^\top\Gamma\hat{V}^\top - V\Gamma^\top\Gamma V^\top\|_F^2 \right] \\
&\leq O\left( \sum_{i=1}^{k} \sum_{j=i+1}^{d} \frac{(\gamma_i^2 - \gamma_j^2)^2}{(\sigma_i - \sigma_j)^2} \right) T \\
&= O\left( \sum_{i=1}^{k} \sum_{j=k+1}^{d} \frac{1}{(\sigma_i - \sigma_j)^2} \right) T \\
&\leq O\left( \sum_{i=1}^{k} \sum_{j=k+1}^{d} \frac{1}{(\sigma_k - \sigma_{k+1})^2} \right) T \\
&\leq O\left( \frac{kd}{(\sigma_k - \sigma_{k+1})^2} T \right)
\end{aligned}
\tag{30}
$$

where the first inequality holds by Theorem 2.2 and the second equality holds in that $\gamma_1 = \cdots = \gamma_k = 1$ and $\gamma_{k+1} = \cdots = \gamma_d = 0$.

By Jensen's Inequality, we have that

$$
\mathbb{E}\left[ \|\hat{V}_k\hat{V}_k^\top - V_kV_k^\top\|_F \right] \leq \sqrt{\mathbb{E}\left[ \|\hat{V}_k\hat{V}_k^\top - V_kV_k^\top\|_F^2 \right]} \leq O\left( \frac{\sqrt{kd}}{(\sigma_k - \sigma_{k+1})} \right) \sqrt{T}.
$$

In the second case, where the singular values of $A$ also satisfy $\sigma_i - \sigma_{i+1} \geq \Omega(\sigma_k - \sigma_{k+1})$ for all $i \leq k$, we have

$$
\begin{aligned}
\mathbb{E}\left[\|\hat{V}_k \hat{V}_k^\top - V_k V_k^\top\|_F^2\right] &= \mathbb{E}\left[\|\hat{V}\Gamma^\top\Gamma\hat{V}^\top - V\Gamma^\top\Gamma V^\top\|_F^2\right] \\
&\leq O\left(\sum_{i=1}^{k}\sum_{j=i+1}^{d} \frac{(\gamma_i^2 - \gamma_j^2)^2}{(\sigma_i - \sigma_j)^2}\right) T \\
&= O\left(\sum_{i=1}^{k}\sum_{j=k+1}^{d} \frac{1}{(\sigma_i - \sigma_j)^2}\right) T \\
&\leq O\left(\sum_{i=1}^{k}\sum_{j=k+1}^{d} \frac{1}{(i-k-1)^2(\sigma_k - \sigma_{k+1})^2}\right) T \\
&\leq O\left(\sum_{i=1}^{k} \frac{d}{(i-k-1)^2(\sigma_k - \sigma_{k+1})^2}\right) T \\
&\leq O\left(\frac{d}{(\sigma_k - \sigma_{k+1})^2}\sum_{i=1}^{k}\frac{1}{i^2}\right) T \\
&\leq O\left(\frac{d}{(\sigma_k - \sigma_{k+1})^2}\right) T
\end{aligned}
\tag{31}
$$

where the first inequality holds by Theorem 2.2 and the second equality holds since $\gamma_1 = \cdots = \gamma_k = 1$ and $\gamma_{k+1} = \cdots = \gamma_d = 0$, the second inequality holds since $\sigma_i - \sigma_{i+1} \geq \Omega(\sigma_k - \sigma_{k+1})$ for all $i \leq k$, and the last inequality holds since $\sum_{i=1}^{k}\frac{1}{i^2} \leq \sum_{i=1}^{\infty}\frac{1}{i^2} = O(1)$.

Thanks to Jensen's Inequality, we have that

$$
\mathbb{E}\left[\|\hat{V}_k \hat{V}_k^\top - V_k V_k^\top\|_F\right] \leq \sqrt{\mathbb{E}\left[\|\hat{V}_k \hat{V}_k^\top - V_k V_k^\top\|_F^2\right]} \leq O(\frac{\sqrt{d}}{(\sigma_k - \sigma_{k+1})})\sqrt{T}.
$$

$\square$

## C  PROOF OF COROLLARY 2.4

*Proof of Corollary 2.4.* We first bound the quantity $\mathbb{E}\left[\|\hat{V}\Sigma_k^\top\Sigma_k\hat{V}^\top - V\Sigma_k^\top\Sigma_k V^\top\|_F\right]$.

Set $\gamma_i = \sigma_i$ for $i \le k$ and $\gamma_i = 0$ for $i > k$. Then by Theorem 2.2 we have

$$\mathbb{E}\left[\|\hat{V}\Sigma_k^\top\Sigma_k\hat{V}^\top - V\Sigma_k^\top\Sigma_k V^\top\|_F^2\right] \le O\left(\sum_{i=1}^{k}\sum_{j=i+1}^{d}\frac{(\gamma_i^2 - \gamma_j^2)^2}{(\sigma_i - \sigma_j)^2}\right)T$$

$$= O\left(\sum_{i=1}^{k-1}\sum_{j=i+1}^{k}\frac{(\sigma_i^2 - \sigma_j^2)^2}{(\sigma_i - \sigma_j)^2} + \sum_{i=1}^{k}\sum_{j=k+1}^{d}\left(\frac{\sigma_i^2 - 0^2}{\sigma_i - \sigma_j}\right)^2\right)T$$

$$= O\left(\sum_{i=1}^{k-1}\sum_{j=i+1}^{k}(\sigma_i + \sigma_j)^2 + \sum_{i=1}^{k}\sum_{j=k+1}^{d}\left(\frac{\sigma_i^2 - \sigma_k^2}{\sigma_i - \sigma_j} + \frac{\sigma_k^2}{\sigma_i - \sigma_j}\right)^2\right)T$$

$$\le O\left(\sum_{i=1}^{k-1}\sum_{j=i+1}^{k}(\sigma_i + \sigma_j)^2 + \sum_{i=1}^{k}\sum_{j=k+1}^{d}\left(\sigma_i + \frac{\sigma_k^2}{\sigma_i - \sigma_j}\right)^2\right)T$$

$$= O\left(\sum_{i=1}^{k-1}\sum_{j=i+1}^{k}(\sigma_i + \sigma_j)^2 + \sum_{i=1}^{k}\sum_{j=k+1}^{d}\left(\sigma_i + \sigma_k\frac{\sigma_k}{\sigma_i - \sigma_j}\right)^2\right)T$$

$$\le O\left(\sum_{i=1}^{k-1}\sum_{j=i+1}^{k}(\sigma_i + \sigma_j)^2 + \sum_{i=1}^{k}\sum_{j=k+1}^{d}\left(\sigma_i + \sigma_k\frac{\sigma_k}{\sigma_k - \sigma_j}\right)^2\right)T$$

$$\le O\left(\sum_{i=1}^{k-1}\sum_{j=i+1}^{k}(\sigma_i + \sigma_j)^2 + \sum_{i=1}^{k}\sum_{j=k+1}^{d}\sigma_i^2 + \sum_{i=1}^{k}\sum_{j=k+1}^{d}\left(\sigma_k\frac{\sigma_k}{\sigma_k - \sigma_j}\right)^2\right)T$$

$$\le O\left(d\|\Sigma_k\|_F^2 + \sum_{i=1}^{k}\sum_{j=k+1}^{d}\left(\sigma_k\frac{\sigma_k}{\sigma_k - \sigma_j}\right)^2\right)T$$

$$\le O\left(d\|\Sigma_k\|_F^2 + k(d-k)\left(\sigma_k\frac{\sigma_k}{\sigma_k - \sigma_{k+1}}\right)^2\right)T. \tag{32}$$

We next bound the quantity $\mathbb{E}\left[\|\hat{V}\hat{\Sigma}_k^\top\hat{\Sigma}_k\hat{V}^\top - \hat{V}\Sigma_k^\top\Sigma_k\hat{V}^\top\|_F\right]$.

Let $E_1$ be the event when $\|G\| > \sqrt{\max(m,d)}\log(1/\delta)$. By Lemma 3.3, we have $\mathbb{P}(E_1) \ge 1 - \delta$. Since $\|\Sigma_k\|_F \le \sqrt{k}\sigma_1$ and $\|\hat{\Sigma}_k\|_F < \sqrt{k}\sigma_1(t)$, we can use the bound

$$\|\Sigma_k^\top\Sigma_k - \hat{\Sigma}_k^\top\hat{\Sigma}_k\|_F < \|\Sigma_k^\top\Sigma_k\|_F + \|\hat{\Sigma}_k^\top\hat{\Sigma}_k\|_F < k\sigma_1 + k\sigma_1^2(t) < 4k\sigma_1^2$$

and hence

$$\mathbb{E}[\|\Sigma_k^\top\Sigma_k - \hat{\Sigma}_k^\top\hat{\Sigma}_k\|_F * 1_{E_1}] < 2\sqrt{k}\sigma_1 * P(E_1) < 4k\sigma_1^2 * \delta.$$

Recall that (from Assumption 2.1) $\delta < \frac{1}{k\sigma_1^2}$. Hence,

$$\mathbb{E}[\|\Sigma_k^\top\Sigma_k - \hat{\Sigma}_k^\top\hat{\Sigma}_k\|_F * 1_{E_1}] < 4$$

Now consider the event $E_1^c$, where $\|G\| < \sqrt{\max(m,d)}\log(1/\delta)$. From above, we have $\mathbb{P}(E_1^c) = 1 - \mathbb{P}(E_1) \le \delta$. For $E_1^c$ we get,

$$\mathbb{E}[\|\Sigma_k^\top\Sigma_k - \hat{\Sigma}_k^\top\hat{\Sigma}_k\|_F * 1_{E_1^c}] < \mathbb{E}[\|(\Sigma_k - \hat{\Sigma}_k)(\Sigma_k + \hat{\Sigma}_k)\|_F * 1_{E_1^c}]$$
$$< \mathbb{E}[\sqrt{T}\|G_k\| * (\|\Sigma_k\|_F + \|\hat{\Sigma}_k\|_F) * 1_{E_1^c}]$$
$$< \mathbb{E}[2\sqrt{k}T\sigma_1\|G_k\| * 1_{E_1^c}]$$
$$< 2\sqrt{k}d\sigma_1\log(1/\delta)\sqrt{T}.$$

Finally, put the two cases together:

$$
\begin{aligned}
\mathbb{E}\left[\|\hat{V}\hat{\Sigma}_k^\top \hat{\Sigma}_k \hat{V}^\top - \hat{V}\Sigma_k^\top \Sigma_k \hat{V}^\top\|_F\right] &= \mathbb{E}[\|\Sigma_k - \hat{\Sigma}_k\|_F] \\
&= \mathbb{E}[\|\Sigma_k - \hat{\Sigma}_k\|_F * 1_{E_1}] + \mathbb{E}[\|\Sigma_k - \hat{\Sigma}_k\|_F * 1_{E_1^c}] \\
&< 4 + 2\sqrt{kd}\sigma_1 \log(1/\delta)\sqrt{T} \\
&= O(\sqrt{kd}\sigma_1 \log(1/\delta))\sqrt{T}.
\end{aligned}
\tag{33}
$$

Combining equation 32 and equation 33, we have

$$
\begin{aligned}
&\mathbb{E}\left[\|\hat{V}\hat{\Sigma}_k^\top \hat{\Sigma}_k \hat{V}^\top - V\Sigma_k^\top \Sigma_k V^\top\|_F\right] \\
&\leq \mathbb{E}\left[\|\hat{V}\hat{\Sigma}_k^\top \hat{\Sigma}_k \hat{V}^\top - \hat{V}\Sigma_k^\top \Sigma_k \hat{V}^\top\|_F\right] + \mathbb{E}\left[\|\hat{V}\Sigma_k^\top \Sigma_k \hat{V}^\top - V\Sigma_k^\top \Sigma_k V^\top\|_F\right] \\
&\leq O\left(\sqrt{d}\|\Sigma_k\|_F + \sqrt{k(d-k)}\left(\sigma_k \frac{\sigma_k}{\sigma_k - \sigma_{k+1}}\right)\right)\sqrt{T} + O(\sqrt{kd}\sigma_1 \log(1/\delta))\sqrt{T} \\
&\leq O\left(\sqrt{d}\|\Sigma_k\|_F + \sqrt{k(d-k)}\left(\sigma_k \frac{\sigma_k}{\sigma_k - \sigma_{k+1}}\right)\right)\sqrt{T}.
\end{aligned}
\tag{34}
$$

$\square$

# D   ADDITIONAL COMPARISONS FOR LOW-RANK COVARIANCE APPROXIMATION

In this section, we present how one can derive high-probability bounds on the quantity $\|\hat{V}\hat{\Sigma}_k^2 \hat{V}^\top - V\Sigma_k^2 V^\top\|_F$ from the subspace perturbation bounds of Davis & Kahan (1970); Wedin (1972) or O'Rourke et al. (2018).

Towards this end, we note that

$$
\|\hat{V}\hat{\Sigma}_k^2 \hat{V}^\top - V\Sigma_k^2 V^\top\|_F \leq \|\hat{V}\hat{\Sigma}_k^2 \hat{V}^\top - \hat{V}\Sigma_k^2 \hat{V}^\top\|_F + \|\hat{V}\Sigma_k^2 \hat{V}^\top - V\Sigma_k^2 V^\top\|_F.
$$

The first term can be bounded as

$$
\|\hat{V}\hat{\Sigma}_k^2 \hat{V}^\top - \hat{V}\Sigma_k^2 \hat{V}^\top\|_F = \|\hat{\Sigma}_k^2 - \Sigma_k^2\|_F = \sum_{i=1}^k \hat{\sigma}_i^2 - \sigma_i^2,
$$

which can be bounded using Weyl's inequality (Lemma 3.2) together with the Gaussian concentration inequality in Lemma 3.3.

For the second term, we have

$$
\begin{aligned}
\|\hat{V}\Sigma_k^2 \hat{V}^\top - V\Sigma_k^2 V^\top\|_F &= \|\hat{V}\Sigma_k^2 \hat{V}^\top - V\Sigma_k^2 V^\top\|_F \\
&= \|\sum_{i=1}^{k-1}(\sigma_i^2 - \sigma_{i+1}^2)(\hat{V}_i \hat{V}_i^\top - V_i V_i^\top) + \sigma_k^2(\hat{V}_k \hat{V}_k^\top - V_k V_k^\top)\|_F \\
&\leq \sum_{i=1}^{k-1}(\sigma_i^2 - \sigma_{i+1}^2)\|\hat{V}_i \hat{V}_i^\top - V_i V_i^\top\|_F + \sigma_k^2\|\hat{V}_k \hat{V}_k^\top - V_k V_k^\top\|_F \\
&\leq \sum_{i=1}^{k-1}(\sigma_i + \sigma_{i+1})(\sigma_i - \sigma_{i+1})\|\hat{V}_i \hat{V}_i^\top - V_i V_i^\top\|_F + \sigma_k^2\|\hat{V}_k \hat{V}_k^\top - V_k V_k^\top\|_F \\
&\leq \sum_{i=1}^{k-1}(\sigma_i - \sigma_{i+1})\frac{\sqrt{i}\sqrt{d}}{\sigma_i - \sigma_{i+1}} + \sigma_k \frac{\sqrt{k}\sqrt{d}}{\sigma_k - \sigma_{k+1}} \\
&= O\left(k^{1.5}\sqrt{d} + \frac{\sigma_k}{\sigma_k - \sigma_{k+1}}\sqrt{k}\sqrt{d}\right).
\end{aligned}
\tag{35}
$$

Plugging into equation 35 the bound of $\|V_i V_i^\top - \hat{V}_i \hat{V}_i^\top\|_F \leq \frac{\sqrt{i}\sqrt{m}}{\sigma_i - \sigma_{i+1}}\sqrt{T}$ w.h.p. implied by Davis & Kahan (1970); Wedin (1972), one has

$$\|\hat{V}\Sigma_k^2\hat{V}^\top - V\Sigma_k^2 V^\top\|_F \leq \sum_{i=1}^{k-1}(\sigma_i + \sigma_{i+1})(\sigma_i - \sigma_{i+1})\frac{\sqrt{i}\sqrt{m}}{\sigma_i - \sigma_{i+1}}\sqrt{T} + \sigma_k^2\frac{\sqrt{k}\sqrt{m}}{\sigma_k - \sigma_{k+1}}\sqrt{T}$$

$$\leq \sqrt{m}\sqrt{T}\sum_{i=1}^{k-1}(\sigma_i + \sigma_{i+1})\sqrt{i} + \sigma_k^2\frac{\sqrt{k}\sqrt{m}}{\sigma_k - \sigma_{k+1}}\sqrt{T}$$

$$\leq 2k^{1.5}\sqrt{m}\sqrt{T}\sigma_1 + \sigma_k^2\frac{\sqrt{k}\sqrt{m}}{\sigma_k - \sigma_{k+1}}\sqrt{T}.$$

One can also instead plug in the bound from Theorem 18 of O'Rourke et al. (2018) (restated here as equation 2 in Section 1.1) into equation 35. When, e.g., $\sigma_k - \sigma_{k+1} \geq \Omega(\max(\sigma_k, \sqrt{m}))$, equation 2 reduces to $\|\hat{V}\Sigma_i^2\hat{V}^\top - V\Sigma_i^2 V^\top\|_F \leq O\left(i\frac{\sqrt{m}}{\sigma_i}\sqrt{T}\right)$ for $i \leq k$ into equation 35. Thus, plugging in this bound into equation 35, one has

$$\|\hat{V}\Sigma_k^2\hat{V}^\top - V\Sigma_k^2 V^\top\|_F \leq \sum_{i=1}^{k-1}(\sigma_i + \sigma_{i+1})(\sigma_i - \sigma_{i+1})i\frac{\sqrt{m}}{\sigma_i}\sqrt{T} + \sigma_k^2 k\frac{\sqrt{m}}{\sigma_k}\sqrt{T}$$

$$\leq O\left(\sum_{i=1}^{k-1}(\sigma_i - \sigma_{i+1})i\sqrt{m}\sqrt{T} + \sigma_k k\sqrt{m}\sqrt{T}\right)$$

$$\leq O\left((\sigma_1 - \sigma_k)k\sqrt{m}\sqrt{T} + \sigma_k k\sqrt{m}\sqrt{T}\right)$$

$$\leq O\left(\sigma_1 k\sqrt{m}\sqrt{T}\right).$$

# E  NUMERICAL SIMULATIONS

In this section, we present numerical simulations that illustrate the theoretical results in Theorem 2.2, and investigate the extent to which the bounds in Theorem 2.2 are tight.

## E.1  SIMULATIONS FOR RANK-$k$ COVARIANCE MATRIX APPROXIMATION

In this set of simulations, we compute the squared Frobenius error for the rank-$k$ covariance approximation problem, $\|\hat{V}\hat{\Sigma}_k^T\hat{\Sigma}_k\hat{V}^T - V\Sigma_k^T\Sigma_k V^T\|_F^2$. We take an input "data" matrix $A$, perturb the matrix by iid Gaussian noise (that is, $\hat{A} = A + \sqrt{T}G$ where $G$ has iid $N(0,1)$ entries), and compute the error $\|\hat{V}\hat{\Sigma}_k^T\hat{\Sigma}_k\hat{V}^T - V\Sigma_k^T\Sigma_k V^T\|_F^2$, for different values of $m, d, k$. In the following simulations we choose the input "data" matrix to be a synthetic data matrix with linearly decaying spectral profile spectral profile $\sigma_i = \sqrt{m} \times (d - i + 1)$ for all $i \in [d]$. We note that, since the noise distribution $G$ is invariant to multiplication orthogonal matrices, we may assume without loss of generality that $A$ is a (rectangular) $m \times d$ diagonal matrix with diagonal entries $\sigma_1, \cdots, \sigma_d$ and zeros in all other entries.

We then plot the ratio of the error observed in the experiments to the r.h.s. of the bound in Cororlary 2.4, $\frac{\|\hat{V}\hat{\Sigma}_k^T\hat{\Sigma}_k\hat{V}^T - V\Sigma_k^T\Sigma_k V^T\|_F^2}{d\|\Sigma_k\|_F^2 + k\sum_{j=k+1}^d(\frac{\sigma_k^2}{\sigma_k - \sigma_j})^2}$, for different values of $m$ (Figure 1), $d$ (Figure 2), and $k$ (Figure 3), keeping the other two variables fixed in each plot.

We observe that, the ratio of the experimentally observed error and our upper bound does not change much (up to a small constant factor) for different values of $m$ or $d$, suggesting that, for matrices $A$ with the above spectral profile, our bound in Corollary 2.4 is tight with respect to $m$ (Figure 1) and $d$ (Figure 2).

On the other hand, we observe (Figure 3) that the ratio of the observed error and our upper bound seems to be smaller for values of $k$ which are far from 1 or $d$, suggesting that Corollary 2.4 may not have a tight dependence on $k$ for input matrices of this spectral profile.

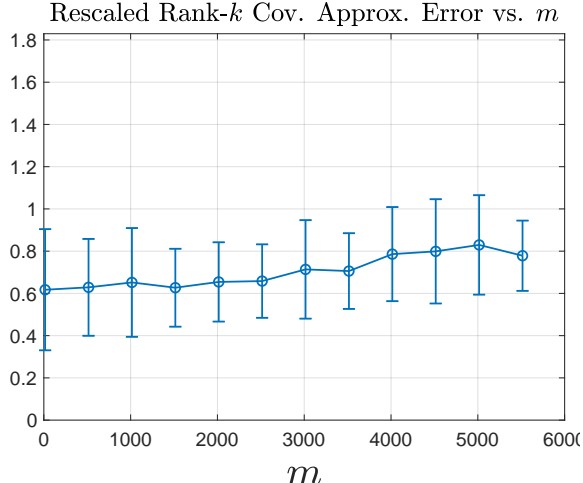

Figure 1: Plot of the rescaled error, that is, the l.h.s. of Corollary 2.4 divided by the r.h.s., $\frac{\|\tilde{V}\hat{\Sigma}_k^T\hat{\Sigma}_k\hat{V}^T - V\Sigma_k^T\Sigma_k V^T\|_F^2}{d\|\Sigma_k\|_F^2 + k\sum_{j=k+1}^d(\frac{\sigma_k^2}{\sigma_k-\sigma_j})^2}$, for different values of $m$. Error bars indicate standard deviation. Here, $d = 15$, $k = 5$, $T = 1$ and the input matrix has spectral profile $\sigma_i = \sqrt{m} \times (d - i + 1)$ for all $i \in [d]$. The rescaled error does not change much, suggesting that, for input matrices of this spectral profile, Corollary 2.4 has a tight dependence on $m$.

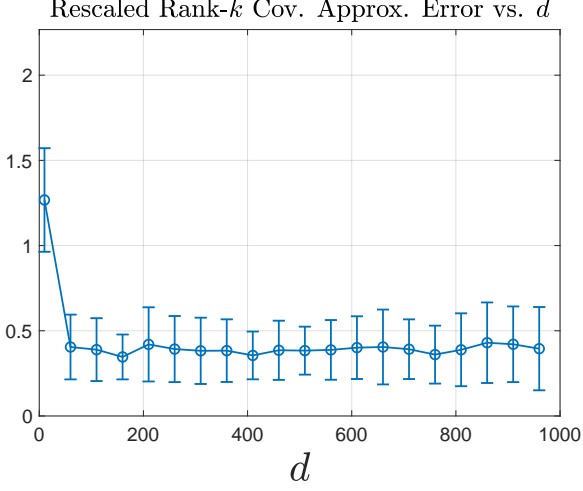

Figure 2: Plot of the rescaled error, that is, the l.h.s. of Corollary 2.4 divided by the r.h.s., $\frac{\|\tilde{V}\hat{\Sigma}_k^T\hat{\Sigma}_k\hat{V}^T - V\Sigma_k^T\Sigma_k V^T\|_F^2}{d\|\Sigma_k\|_F^2 + k\sum_{j=k+1}^d(\frac{\sigma_k^2}{\sigma_k-\sigma_j})^2}$, for different values of $d$. Error bars indicate standard deviation. Here, $m = 1000$, $k = 5$, $T = 1$ and the input matrix has spectral profile $\sigma_i = \sqrt{m} \times (d - i + 1)$ for all $i \in [d]$. The rescaled error does not change much, suggesting that, for input matrices of this spectral profile, Corollary 2.4 has a tight dependence on $d$.

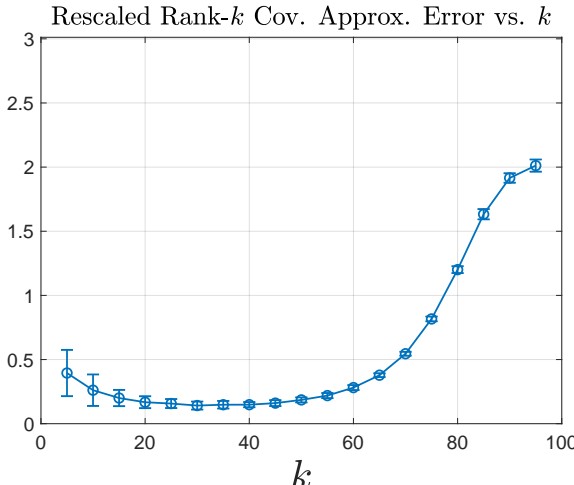

Figure 3: Plot of the rescaled error, that is, the l.h.s. of Corollary 2.4 divided by the r.h.s., $\frac{\|\hat{V}\hat{\Sigma}_k^T\hat{\Sigma}_k\hat{V}^T - V\Sigma_k^T\Sigma_k V^T\|_F^2}{d\|\Sigma_k\|_F^2 + k\sum_{j=k+1}^d (\frac{\sigma_k^2}{\sigma_k-\sigma_j})^2}$, for different values of $d$. Error bars indicate standard deviation. Here, $m = 1000$, $d = 5$, $T = 1$ and the input matrix has spectral profile $\sigma_i = \sqrt{m} \times (d - i + 1)$ for all $i \in [d]$. The rescaled error seems to be smaller for values of $k$ which are far from 1 or $d$, suggesting that Corollary 2.4 may not have a tight dependence on $k$ for input matrices of this spectral profile.

### E.2 SIMULATIONS FOR SUBSPACE RECOVERY

In this section, we present numerical simulations that illustrate the theoretical results in Theorem 2.2, and investigate the extent to which the bounds in Theorem 2.2 are tight.

### E.3 SIMULATIONS FOR RANK-$k$ COVARIANCE MATRIX APPROXIMATION

In this set of simulations, we compute the Frobenius norm error for the subspace recovery problem, $\|\hat{V}\hat{V}^T - VV^T\|_F$. We take an input "data" matrix $A$, perturb the matrix by iid Gaussian noise (that is, $\hat{A} = A + \sqrt{T}G$ where $G$ has iid $N(0,1)$ entries), and compute the error $\|\hat{V}\hat{V}^T - VV^T\|_F$, for different values of $m, d, k$. As in the simulations of Section E.1, we choose the input "data" matrix to be a synthetic data matrix with linearly decaying spectral profile spectral profile $\sigma_i = \sqrt{m} \times (d-i+1)$ for all $i \in [d]$.

We then plot the ratio of the error observed in the experiments to the r.h.s. of the bound in Corollary 2.3, $\frac{\|\hat{V}_k\hat{V}_k^T - V_k V_k^T\|_F}{\sqrt{d}/(\sigma_k-\sigma_{k+1})}$, for different values of $m$ (Figure 4), $d$ (Figure 5), and $k$ (Figure 6), keeping the other two variables fixed in each plot.

We observe that, the ratio of the experimentally observed error and our upper bound does not change much (up to a small constant factor) for different values of $m$ or $k$, suggesting that, for matrices $A$ with the above spectral profile, our bound in Corollary 2.3 is tight with respect to $m$ (Figure 4) and $k$ (Figure 6).

On the other hand, we observe (Figure 5) that the ratio of the observed error and our upper bound seems to decrease with $d$, suggesting that, Corollary 2.3 may not be tight in $d$ for input matrices of this spectral profile..

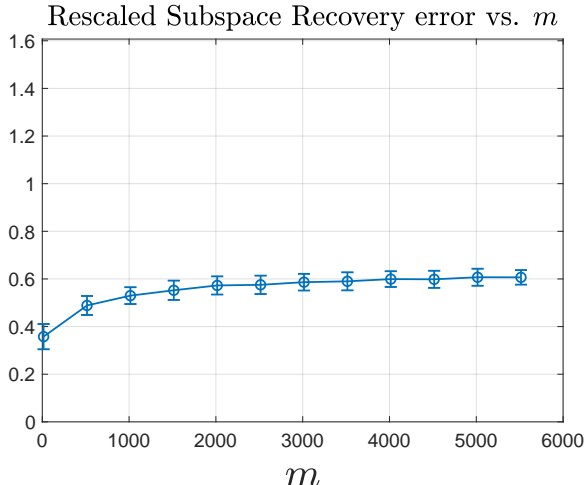

Figure 4: Plot of the rescaled subspace recovery error, that is, the l.h.s. of Corollary 2.3 divided by the r.h.s., $\frac{\|\hat{V}_k \hat{V}_k^\top - V_k V_k^\top\|_F}{\sqrt{d}/(\sigma_k - \sigma_{k+1})}$, for different values of $m$. Error bars indicate standard deviation. Here, $d = 15$, $k = 5$, $T = 1$ and the input matrix has spectral profile $\sigma_i = \sqrt{m} \times (d - i + 1)$ for all $i \in [d]$. The rescaled error does not change much, suggesting that, for input matrices of this spectral profile, Corollary 2.3 has a tight dependence on $m$.

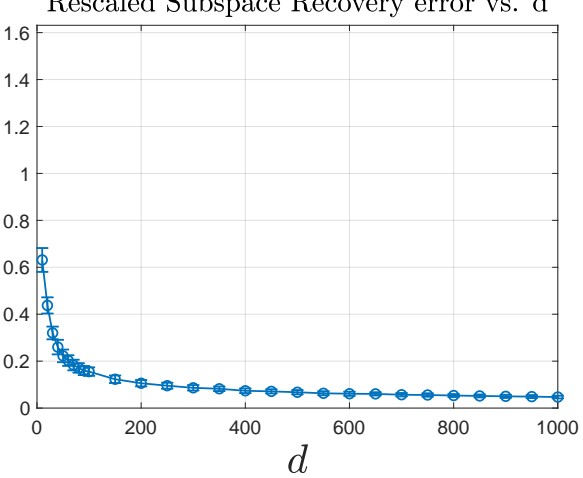

Figure 5: Plot of the rescaled subspace recovery error, that is, the l.h.s. of Corollary 2.3 divided by the r.h.s., $\frac{\|\hat{V}_k \hat{V}_k^\top - V_k V_k^\top\|_F}{\sqrt{d}/(\sigma_k - \sigma_{k+1})}$, for different values of $m$. Error bars indicate standard deviation. Here, $d = 15$, $k = 5$, $T = 1$ and the input matrix has spectral profile $\sigma_i = \sqrt{m} \times (d - i + 1)$ for all $i \in [d]$. The rescaled error decreases with $d$, suggesting that, Corollary 2.3 may not be tight in $d$ for input matrices of this spectral profile.

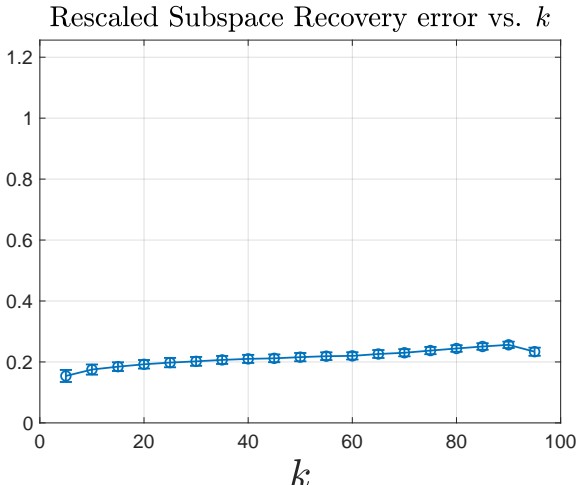

Figure 6: Plot of the rescaled subspace recovery error, that is, the l.h.s. of Corollary 2.3 divided by the r.h.s., $\frac{\|\hat{V}_k\hat{V}_k^\top - V_k V_k^\top\|_F}{\sqrt{d}/(\sigma_k - \sigma_{k+1})}$, for different values of $k$. Error bars indicate standard deviation. Here, $d = 15$, $k = 5$, $T = 1$ and the input matrix has spectral profile $\sigma_i = \sqrt{m} \times (d - i + 1)$ for all $i \in [d]$. The rescaled error does not change much, suggesting that, for input matrices of this spectral profile, Corollary 2.3 has a tight dependence on $k$.

