# OpenReview forum: "Singular Subspace Perturbation Bounds via Rectangular Random Matrix Diffusions"
_ICLR.cc/2025/Conference — ICLR 2025 Poster_

### Official Review · Reviewer_Bu3u · 2024-11-04

**Soundness:** 3
**Presentation:** 3
**Contribution:** 1
**Rating:** 3
**Confidence:** 5

**Summary:**

This paper considers estimation of the right singular subspace of a matrix $A\in\mathbb{R}^{m\times d}$ when the matrix is perturbed by a Gaussian random matrix. The authors focused on the case $d>m$ and derived a perturbation Frobenius bound independent of $m$.

**Strengths:**

- The paper is easy to read.
- The problem considered is important and has wide applications to many machine learning problems.

**Weaknesses:**

The main weakness of this paper is that the question posed in the Introduction has already been addressed in the existing literature. The prior works have already proposed minimax optimal subspace estimators that attain tighter bounds than the one derived in this paper. Moreover, the required signal-to-noise condition in this paper is also more stringent than those in the prior works.

In [1,2,3], the authors proposed left singular subspace estimators of a matrix $A\in\mathbb{R}^{d_1\times d_2}$, which achieve the statistically optimal estimation bounds (in both spectral norm and $({2,\infty})$ bound) that only depend on $d_1$ when the noise variance $\sigma^2$ is small. In addition, it has been shown that when $\sigma^2$ is large, it is information-theoretically impossible to obtain an estimation bound that is independent of $d_2$.

By transposing the target matrix and converting an $({2,\infty})$ bound to a Frobiuns bound (multiplying by $\sqrt{d}$), the estimation guarantee for the right singular subspace in terms of the Frobenius norm follows directly from these prior results. Further, the signal-to-noise ratio condition in this paper $\sqrt{T}/(\sigma_1-\sigma_{i+1}) = \tilde{O}(1/\sqrt{m})$ (Assumption 2.1) is much stringent than the conditions required in these existing works ($\sqrt{T}/(\sigma_1-\sigma_{i+1}) = \tilde{O}(1/\sqrt{d} \wedge 1/(md)^{1/4})$).  Moreover, the estimation guarantees in the prior work are established in high probability while this paper only establishes it in expectation.

In short, under less stringent signal-to-noise ratio conditions, prior works have already achieved statistically better bounds (which are indeed minimax optimal up to log factors) for the singular subspace than the bound obtained in this paper.

Therefore, the intellectual contribution of this paper is limited. Given the estimation error in the existing works is already minimax optimal and the required SNR condition therein is shown to be necessary to ensure consistent estimation, one approach to improve the contribution might be removing the log factors in the error bounds.


[1] "Subspace Estimation from Unbalanced and Incomplete Data Matrices: $\ell_{2,\infty}$ Statistical Guarantees," Annals of Statistics, 944-967, 2021
[2] "Inference for heteroskedastic PCA with missing data," Annals of Statistics, 729-756, 2024
[3] "Deflated HeteroPCA: Overcoming the Curse of Ill-Conditioning in Heteroskedastic PCA," Annals of Statistics, to appear

**Questions:**

N/A

---

### Official Review · Reviewer_JJsP · 2024-11-04

**Soundness:** 3
**Presentation:** 3
**Contribution:** 3
**Rating:** 5
**Confidence:** 3

**Summary:**

The paper studies a very interesting problem, namely that of bounding the Frobenius norm of the subspaces spanned by the top-k singular vectors of any fixed matrix A and A+G. The paper is well structured and reads well, but has very few thematic overlaps with the ICLR.

I am rejecting this paper for two reasons. First, the authors used the wrong template with significantly wider margins than the standard ICLR template. While this is nitpicking, choosing the right template comes with no overhead but is critical to maintain fair standards on the length of the paper among all submissions.

The second reason is that the paper targets a problem in random matrix theory, which would fit well to a conference like ALT or COLT or a mathematical journal. In order to submit this paper to ICLR, I would expect at least one longer section highlighting the connections to machine learning and demonstrate how these bounds help us to arrive at non-trivial novel insights directly relevant for machine learning.

**Strengths:**

I find the topic very interesting. Proving these results is non-trivial and insightful.

**Weaknesses:**

-

**Questions:**

-

**Details Of Ethics Concerns:**

-

---

### Official Review · Reviewer_6eUD · 2024-11-05

**Soundness:** 3
**Presentation:** 3
**Contribution:** 3
**Rating:** 6
**Confidence:** 3

**Summary:**

This paper studies the Frobenius norm distance between the projection matrix to the top rank-k row space of A and the projection matrix to the top rank-k row space of A + G where G is a random Gaussian matrix and A is a tall n x d matrix, i.e., n >> d. The previous result by O’Rourke et al. (2023) shows that if one considers both distances between the projection matrices corresponding to column space and row space, then they can get a tight result. In this work, authors gave an improved bound when considering the projection matrices to row spaces only.

**Strengths:**

1. The main proof technique requires a clever combination of many building blocks: They firstly regard the added Gaussian noise G as a Brownian motion. This Brownian motion induces a stochastic diffusion process on the singular values and singular vectors. The evolution of these eigenvalues and eigenvectors is determined by a system of stochastic differential equations. Then, one can apply Ito’s lemma from stochastic calculus to track and analyze the Frobenius norm distances between the top-k projection matrices. It requires solid work to make the entire theoretical analysis work.

1. The problem is very fundamental, and the improvement of the bound is by a factor of sqrt(n / d), when n >> d, this improvement is very significant.

**Weaknesses:**

1. As mentioned in the conclusion, the result needs to assume a reasonable top-k singular value gap of A.

2. A bigger concern would be the interest of the audience of the venue. Although there are some potential applications of the derived bound for handling data corruption / noise, or differential privacy (Gaussian mechanism), there is no concrete application discussed in the paper.

**Questions:**

In my opinion, the proposed bound may be used to improve the tradeoffs between privacy and accuracy of some differentially private algorithms in numerical linear algebra. Can authors show some with provable guarantees? Or can authors give any concrete applications of the improved bound in the machine learning literature?

---

### Official Review · Reviewer_PBwp · 2024-11-07

**Soundness:** 3
**Presentation:** 3
**Contribution:** 3
**Rating:** 8
**Confidence:** 2

**Summary:**

This paper presents new bounds on the Frobenius norm of the perturbation to the singular subspace spanned by the top-k right singular vectors of a matrix $A \in \mathbb{R}^{m \times d}$, when $A$ is perturbed by a matrix $G \in \mathbb{R}^{m \times d}$ with Gaussian $\mathcal{N}(0,T)$ random entries. More specifically, assuming $A$ has a top-k singular value gap of at least $\Omega(\sqrt{mT})$ (i.e. $\sigma_i-\sigma_{i+1} \geq \Omega(\sqrt{mT}) $ for all $i \in [k]$), the paper bounds the expected Frobenius norm distance between the projection matrices on the top $k$ right singular subspaces of $A$ and $A+G$ by roughly $O( \frac{ \sqrt{kdT} }{ \sigma_k - \sigma_{k+1} })$ where $\sigma_1 \geq \ldots \geq \sigma_d \geq 0$ are the singular values of $A$. When $ \sigma_i  - \sigma_{i+1} \geq \Omega( \sigma_k -\sigma_{k+1} )$ for $i \in [k]$ is also true, the bound is $O(\frac{ \sqrt{dT} }{ \sigma_k - \sigma_{k+1}  })$ which improves upon previously known high probability bounds on the actual Frobenius norm distance by $\frac{\sqrt{m}}{\sqrt{d}}$. The paper also improves previously known bounds for the rank-k covariance matrix approximation by a factor of $\frac{\sqrt{m}}{\sqrt{d}}$.

To derive the bounds, the paper views the addition of a gaussian noise matrix to $A$ as a Brownian motion on the entries of a matrix $\Phi(t) := A+B(t)$ where $B(t)$ is a matrix whose entries undergo standard Brownian motion. This Brownian motion induces a stochastic diffusion process, known as the Dyson-Bessel process on the singular values and singular vectors of $\Phi(t)$. The evolution of the eigenvalues and eigenvectors of of this matrix is determined by a system of stochastic differential equations. Using techniques from stochastic calculus (Ito's lemma), the evolution of the Frobenius distance can be tracked as a stochastic integral of a sum-of-squares of perturbations to the right singular vectors of $\Phi(t)$. Overall, higher order matrix derivatives which appear in the Taylor series expansion of deterministic perturbations vanish in the stochastic case due to the independence of random noise in the Brownian motion and this gives the stronger bounds.

**Strengths:**

Though someone with a more thorough grasp of stochastic calculus would be much better suited to comment on the merit of the exact proof technique used in the paper, I found the overall technique of viewing the noise addition as Brownian motion and then using stochastic differential equations to track the eigenspace evolution quite novel.

The bounds on the Frobenius norm distance on the right singular subspace presented have an improved dependence of $\sqrt{d}$ instead of $\sqrt{m}$ and in many applications where we have $m>>d$, I think the bounds could be useful (when there are large enough singular value gaps).

That being said, I have a few concerns about the paper which I list below.

**Weaknesses:**

Though this is not a weakness per se, I'm not sure how suited this paper is for ICLR which is primarily an applied ML venue. I feel at least a core theoretical venue like COLT or maybe a mathematics journal is better suited for this kind of work. That being said, if the authors are able to show a concrete ML related application of the utility of their bounds, that would make the paper a stronger contender for publication at ICLR and also improve the overall quality of the paper. For example, as mentioned in the paper, in differential privacy, gaussian noise is added to the data matrix to preserve privacy (Dwork et al 2014). Can the authors try to use their bounds to see if they can get any improved bounds in this case?

Also, the bounds in previous papers like O'Rourke et al. 2023 are high probability bounds while here, the bounds are just on the expected Forbenius norm. How difficult would it be to use the same techniques to get high probability bounds?

I also have a couple of questions about comparison to previous work (specially to O'Rourke et al. 2023) which I list below in the questions section.

**Questions:**

I'm confused about the claim in the paper that Theorem 7 of O'Rourke et al. 2023 achieves a bound of $O(\sqrt{mk}/\sigma_k)$.  It seems that in Theorem 7 they assume that $ \sigma_i  - \sigma_{i+1}  \geq r^2 $ for $ i \in [k]$ (where $r$ is the rank of A). I maybe misunderstanding something but it seems that eq 8 of Theorem 7 then gives us an upper bound of $\frac{\sqrt{k}}{r}+ \frac{|| E ||_2}{\sigma_k} \leq \frac{\sqrt{k}}{r} +\frac{\sqrt{m}}{\sigma_k}$ ? Can the authors please clarify this?

Can the authors also discuss the assumptions used in O'Rourke et al. 2023 or other previous works to derive the bounds and how it compares to their own on the singular values gaps i.e.  $ \sigma_i  - \sigma_{i+1} \geq \Omega( \sigma_k -\sigma_{k+1} )$? This would be helpful in doing a more fair comparison of the bounds.

Though this is a very interesting and can definitely be published at ICLR, I would request other reviewers and the area chair to the issues I have raised above. I'm giving a marginal accept. Assuming the points I have raised above are resolved, I will be happy to give a stronger accept.

---

### Official Review · Reviewer_vMhC · 2024-11-11

**Soundness:** 3
**Presentation:** 2
**Contribution:** 2
**Rating:** 5
**Confidence:** 3

**Summary:**

The paper provides a new upper-bound for the right singular vectors of a perturbed matrix by Gaussian noise. The authors show that under a set of extra assumptions compared to the literature, we are able to further reduce the bound for tall matrices. The authors used the same technique as Manghoubi et al. 2022 for using Dyson Brownian motions to analyze the singular values and vectors of the perturbed matrix.

**Strengths:**

The paper is concise and with a well-defined objective of finding the upper-bound on the error of singular vectors of perturbed matrices. The strength of the paper is therefore on the theoretical results.

**Weaknesses:**

Although I enjoyed reading the manuscript, I should mentioned as an ICLR submission, it is strangely formed. It is a very short introduction on the problem and the objective of the paper is not well-motivated. Furthermore, although abstractly mentioned that the results can be used in differentially private subspace reconstruction, there is neither further analysis on that (which is fairly easy to do) nor any experiments, including experiments that bridge the work to the machine learning / privacy area.

Furthermore, reading the paper and the abstract, it appears that the manuscript is written in a rush and with a variety of notational issues, brevity of explanations, etc. I mention such issues in Questions section.

Finally, although the manuscript uses a similar approach as Manghoubi et al. 2022, it appears that they achieve a tighter bounds. This would ask for further missing explanation on what is the novelty of this paper compared to the literature and how is that making this new bound possible. In the next step, the authors are expected to show that how this reduction of the bound is not an incremental result and how it improves the methods that need such upper-bounds (e.g., DP-subspace-recovery) in real applications.

**Questions:**

1- Could you please motivate the theoretical result with some examples, even simply using toy datasets on differential privacy?

2- Could you run a set of experiments on some datasets or even synthetic matrices that show the upper-bound and compares it to the real error?

2- Could you further motivate why tall matrices are of interest in this particular setting?

----
Technical Comments:

1- In Line 144, the authors referred to Appendix J of. Manghoubi et al. to justify the assumption of exponential decay. I have checked and their manuscript has no Appendix J. Could you clearly mention their result in the manuscript and correctly refer to their work?

2- Line 191 has no period in the end.

3- In Line 204, should it be $\gamma_i=0$ for $i>k$?

4- In scientific writing, when we use a certain theorem, equation, etc. we refer to them as first name and last name and therefore capitalize "Theorem" and "Equation". Could you further edit this on the manuscript in Line 292, 343, 355, etc.

5- The factor 64 and 32 in Eq. (14) is missing in Eq. (12).

6- In Line 393, the authors write "Noting that the second term on the right-hand side of (12) is at least as small as the first term". Could you explain why this is the case?

7- In Line 595 please explain why $\mathrm{d} v_i\mathrm{d} v_i^T$ is neglected.

8- Line 649 is not clear to me and the identities mentioned in 658-661 is not helping either. Please rewrite this section of the proof and explain why these identites and inequalities hold.

9- The proof of Lemma A.3, although seems very rudimentary and practical for the rest of the proofs, it is not clearly written. If we assume $|\sigma_i(t)-\sigma_i|\leq \sigma_i + \|G\|_2$, then the triangle inequality concludes $|\sigma_i(t)-\sigma_j(t)|=|\sigma_i(t)-\sigma_i +\sigma_j-\sigma_j(t) +\sigma_i - \sigma_j|\geq |\sigma_i - \sigma_j| - |\sigma_i(t)-\sigma_i +\sigma_j-\sigma_j(t) +\sigma_i | \geq |\sigma_i - \sigma_j| - \sigma_i-\sigma_j-4\sqrt{m}\log 1/\delta$, which is not what the authors claim. From what I understand the singular values of a perturbed matrix has the error bounded by the norm of that matrix. Therefore, $|\sigma_i(t)-\sigma_i|\leq \|G\|_2$ instead of $|\sigma_i(t)-\sigma_i|\leq \sigma_i + \|G\|_2$? Also, how does $T$ not appear in the inequality?

10- In Line 698, factor 2 is not needed since $\sigma_i(t)\geq 0$ and $\sqrt{a^2+b^2}\leq a+b$?

11- In 721 and 725, why there is factor 3? Frobenius is a norm and therefore has triangle inequality.

12- In Line 752, please either stick with the notation AA^T or instead use $\langle\rangle$ instead of $<>$.

13- From where does factor 2 in 772 appears?

14- In 775, should we have $4\mathrm{d}t$?

---

### Meta-Review · Area_Chair_qUSz · 2024-12-21

**Metareview:**

This paper provides new matrix perturbation bounds for the top singular vectors of a matrix A under Gaussian perturbations. The authors improve prior results in the setting when the matrix is much taller than it is wide (a common setting in data applications). Most of the reviewers (besides one outlier) felt that the work was interesting and addresses a fundamental question. There was a broad concern about fit to ICLR given the lack of applications -- the work might find a better home in a theory-focused conference. To their credit, the authors added a section during the rebuttal phase on an application to differential privacy (basically a direct corollary of their result is they can bound the accuracy of the top subspace of a data matrix perturbed using the common Gaussian mechanism. Nevertheless, on balance, this paper seems like a nice contribution to include in ICLR.

**Additional Comments On Reviewer Discussion:**

During the rebuttal phase, the authors responded to some seeming misunderstandings by Reviewer Bu3u. While the reviewer did not follow-up on the final response, due to the authors response, I am discounting the low score of this reviewer in my decision, which I believe is based on an incorrect belief on how the authors results relate to prior work.

---

### Decision · Program_Chairs · 2025-01-22

Accept (Poster)